# Biochemical principles of miRNA targeting in flies

**Joel Vega-Badillo** [1] ✉, **Phillip D. Zamore** [1,2] **& Karina Jouravleva** [3] ✉

MicroRNAs direct Argonaute proteins to repress complementary target mRNAs via mRNA degradation or translational inhibition. While mammalian miRNA targeting has been well studied, the principles by which *Drosophila* miRNAs bind their target RNAs remain to be fully characterized. Here, we use RNA Bind-n-Seq to systematically identify binding sites and measure their affinities for five highly expressed *Drosophila* miRNAs. Our results reveal a narrower range of binding site diversity in flies compared to mammals, with fly miRNAs favoring canonical seed-matched sites and exhibiting limited tolerance for imperfections within these sites. We also identified non-canonical site types, including nucleation-bulged and 3'-only sites, whose binding affinities are comparable to canonical sites. These findings establish a foundation for future computational models of *Drosophila* miRNA targeting, enabling predictions of regulatory outcomes in response to cellular signals, and advancing our understanding of miRNA-mediated regulation in flies.

In plants and animals, ~22-nt microRNAs (miRNAs) guide AGO-clade Argonaute proteins to repress partially complementary mRNA targets by accelerating their degradation[1–3] and inhibiting their translation[4–8], with recent mechanistic insights indicating that these effects may occur in parallel rather than as a sequential process[9]. Many loss-of-function miRNA mutants display developmental, physiological, and behavioral defects[10,11].

In animals, the miRNA seed (miRNA nucleotide positions g2–g7) is the primary determinant for targeting, and pairing to the miRNA seed often suffices for target binding and repression[6,12–16]. Additional pairing to the miRNA 3' end (centered around g13–g16) can reinforce recognition of seed-matched targets and compensate for weak or imperfect seed pairing[16–20]. However, such 3'-supplementary or compensatory sites appear to be rare[16,21,22]. Mammalian miRNA binding tolerates single wobble, bulged or mismatched nucleotides at specific positions within the seed[23–25]. Adding to this complexity, some miRNAs bind "central sites" sites bearing extensive complementarity to the miRNA center (nt 4–14 or 5–15) or 3' nucleotides ("3'-only sites") with affinities comparable to canonical seed-matched sites[24–26].

Partial complementarity of miRNA targets, non-canonical binding modes, local sequence-dependent interactions, and the generally modest repression of targets make prediction of animal miRNA regulatory targets difficult, despite two decades of research[24,27]. The most successful computational approaches typically predict miRNA binding rather than biologically important regulation[11,24,27]. In contrast to the large number of targets uncovered by cell culture or computational experiments[18,21,22,28,29], genetic experiments in *Caenorhabditis elegans*, *Drosophila melanogaster*, and mice suggest that miRNAs can act as 'master regulators' of specific biological processes, with repression of as few as a single mRNA explaining the primary function of the miRNA. In *Caenorhabditis elegans*, *lin-4* represses *lin-14* and is essential for transition between larval stages[30]. In *Drosophila melanogaster*, *bantam* represses the pro-apoptotic gene *head involution defective* (*hid*), regulating tissue growth in post-embryonic development[31,32]. In circadian cells, *bantam* silences a core circadian clock gene (*clock*) affecting circadian timekeeping[33]. Transgenic flies expressing *clock* without 3' UTR *bantam* sites are arrhythmic; introduction of a wild-type copy of *clock* rescues this rhythm defect. In mice, miR-9 represses the orphan nuclear receptor tailless homolog (TLX) and induces neuronal differentiation[34]. Neural stem cells expressing TLX lacking miR-9 binding sites recover proliferative capacity and are not able to differentiate into astrocytes.

[1]RNA Therapeutics Institute, University of Massachusetts Chan Medical School, Worcester, MA, USA. [2]Howard Hughes Medical Institute, University of Massachusetts Chan Medical School, Worcester, MA, USA. [3]Laboratoire de Biologie et Modélisation de la Cellule, École Normale Supérieure de Lyon, CNRS UMR5239, Inserm U1293, Université Claude Bernard Lyon 1, Lyon, France. ✉e-mail: joel.vegabadillo@umassmed.edu; karina.jouravleva@ens-lyon.fr

Much of our understanding of miRNA targeting derives from high-throughput biochemical analyses of human and mouse miRNAs[23–25,35], and these insights have improved miRNA target prediction[24]. Here, we use RNA Bind-n-Seq (RBNS) to identify binding sites and obtain high-throughput affinity measurements for five abundant *Drosophila* miR-NAs loaded into Ago1. Analyses of these data reveal that fly miRNAs bind a more restricted number of site types compared to mammals. Our experiments identify the sequence determinants for fly miRNA binding, which promise to spur the development of computational tools to accurately predict miRNA-mediated silencing in flies.

## Results

### Absolute equilibrium dissociation constants can be estimated from RBNS data

RNA bind-n-seq (RBNS) is a high-throughput sequencing method developed to study the affinity and specificity of RNA-binding proteins[36]. When combined with maximum-likelihood analysis, RBNS recapitulates equilibrium dissociation constants ($K_D$) for simulated data[25], and miRNAs[25] and piRNAs[37] whose affinities have been determined by ensemble or single-molecule methods. To further test the validity of this approach, we applied the maximum-likelihood strategy to published RBNS data for human PUM2[38] to obtain $K_D = 0.12 \pm 0.01$ nM for the PUM consensus binding site, UGUAUAUA (Supplementary Data 1). This result agrees well with the reported $K_D$ values for human PUM2, 0.17 nM[39], fly Pumilio, $0.42 \pm 0.07$ nM[40], and *Saccharomyces cerevisiae* PUF4, 0.88 nM[41].

### Canonical binding can be predicted by free energy of base-pairing

miRNAs display sequence-specific differences in binding their canonical sites: those with strong predicted free energy of site pairing bind their target sites with higher affinity than miRNAs with weak seed-pairing[23–25]. Nevertheless, the difference in binding affinities of mammalian miRNAs is less than might have been expected. To study the energetics of canonical binding of fly miRNAs, we loaded recombinant Ago1 with one of five different miRNAs—*let-7*, *bantam*, miR-184, miR-11, and miR-124—chosen because they play key roles in *Drosophila* development and homeostasis[15,16,32,42–50] and span a range of GC content within their seeds and 3′ regions (Figs. 1a and S1). In agreement with previous studies in animals, all five miRNAs bound 8mer sites (targets with complementarity to positions g2–g8 and an adenosine opposite miRNA g1, i.e., t1A) with the highest affinity: $K_D = 4.54$ pM (95% CI = [4.42, 4.64]) for *let-7* to $K_D = 29.9$ pM (95% CI = [29.2, 30.6]) for *bantam* (Fig. 1b and Supplementary Data 1). The hierarchy of other canonical sites varied with the miRNA sequence. For *let-7*, miR-184 and miR-124, 7mer-m8 displayed a ~4-fold higher affinity than 7mer-A1; but for *bantam* and miR-11, the dissociation constants of 7mer-m8 and 7mer-A1 were almost identical. The ranking of 6mer, 6mer-m8 and 6mer-A1 was also miRNA-specific: $K_D^{6mer} < K_D^{6mer\text{-}m8} < K_D^{6mer\text{-}A1}$ for miR-184 but $K_D^{6mer} < K_D^{6mer\text{-}A1} < K_D^{6mer\text{-}m8}$ for *let-7*; for miR-11, the affinities of these three site types were indistinguishable. These results mirror those for mammalian miRNAs[24,25]. Moreover, direct binding measurements found no substantive difference in affinity between a seed-matching ($K_D = 5 \pm 3$ pM) and a fully complementary target ($K_D = 2 \pm 1$ pM; Fig. 1c), suggesting that seed complementarity dominates *Drosophila* Ago1 target binding, just like mammalian AGO2[23,24,51–53] (Table 1).

### Ago1 does not tolerate G:U pairs within the seed

*Drosophila* miRNAs have been proposed to repress mRNAs containing seed-matched binding sites interrupted by G:U wobble-pairs[14,32]. However, these mRNAs also contain canonical binding sites free of G:U wobble pairs. Our RBNS experiments do not support the view that fly Ago1 tolerates G:U wobble pairs in the seed sequence. We measured the binding affinity of 8mer and 7mer-m8 binding sites containing one, two or three G:U wobble-pairs (Figs. 2a and S2a). A single G:U base-pair at

positions t2–t7 reduced the binding affinity of 8mer and 7mer-m8 sites by at least 6-fold. Binding sites with G:U wobble-pairing at the last position of a site type had dissociation constants comparable to the corresponding shorter site type: $K_{D,8mer\text{-}w8}^{let\text{-}7} = 15.4$ pM vs. $K_{D,7mer\text{-}A1}^{let\text{-}7} = 32.4$ pM, $K_{D,8mer\text{-}w8}^{miR\text{-}11} = 61.2$ pM vs. $K_{D,7mer\text{-}A1}^{miR\text{-}11} = 91.7$ pM, $K_{D,7mer\text{-}m8,w2}^{let\text{-}7} = 278$ pM vs. $K_{D,6mer\text{-}m8}^{let\text{-}7} = 439$ pM, $K_{D,7mer\text{-}m8,w2}^{miR\text{-}184} = 154$ pM vs. $K_{D,6mer\text{-}m8}^{miR\text{-}184} = 409$ pM, and $K_{D,7mer\text{-}m8,w2}^{bantam} = 866$ pM vs. $K_{D,6mer\text{-}m8}^{bantam} = 1356$ pM. Sites with >1 G:U had $K_D$ values comparable to RNA containing no identifiable sites (Fig. S2b). That is, binding to targets with >1 G:U is indistinguishable from background (Figs. 2a and S2a). We conclude that G:U wobbles within the seed region are detrimental for Ago1 binding.

### Additional 3′ pairing compensates for weak seed complementary

Validated targets of *bantam*, miR-2a, and miR-7 contain compensatory sites: 8mer sites with a single G:U pair compensated by a strong pairing to the miRNA 3′ end conferred repression in vivo[14,16]. RBNS is limited to studies of motifs ≤12-nt long, as longer motifs are poorly represented in the sequencing data[24,25]. We designed a *let-7* target RNA fully complementary to *let-7* but for a single G:U at g4, the position at which a wobble pair is most detrimental to seed binding (Fig. 2b, middle panel), and measured its binding affinity in a competition assay with a ³²P-radiolabeled target complementary to the entirety of *let-7*. The target bearing a single G:U pair displayed nearly the same affinity as the fully complementary target. In contrast, introduction of a second G:U pair at position g5 decreased its binding affinity 120-fold (Fig. 2b, right panel). We obtained similar results for binding sites bearing mismatched nucleotides instead of G:U pairs (Fig. 2c). Together, our data indicate that compensation by 3′ pairing can rescue sites with a single G:U or mismatch in the miRNA seed.

### The effect of target mismatches differs across the seed

The seed region of miRNAs behaves as an RNA helix—mismatches in the center have the greatest effect, because they disrupt coaxial stacking more than mismatches at the extremities[23,25,51]. Consistent with this idea, single mismatches to seed positions g3, g4, and g5 led to the largest reduction in binding for *Drosophila* Ago1 (Figs. 3a and S3a). At these positions mismatched nucleotides A and G are the least tolerated in half of the occurrences, likely because their purine rings are larger than pyrimidines and thereby cause more steric hindrance.

Interestingly, miR-184-guided Ago1 displays an unusual behavior in binding mismatched targets. Like *let-7*, *bantam*, and miR-11, it binds 8mer targets with single mismatches at positions t4 and t5 less tightly than fully matched sites (17–69-fold increase in $K_D$ values). But its affinities for t4- and t5-mismatched targets were ≤200 pM, unexpectedly high compared to 8mer sites bearing a single mismatch at t3 or t6 (Fig. 3a). As expected, sites with t4 or t5 mismatched nucleotides have higher predicted $\Delta G$ values than sites with t3 or t6 mismatches (Fig. 3b) and are not sequestered in stable secondary structures occluding their binding (Fig. 3c). Our data suggest that Ago1 may bind sites containing one single central mismatch but identifying universal rules for predicting which miRNAs can bind 8mer-x4R sites with unexpectedly high affinity will require high-throughput target binding measurements for a larger set of miRNAs.

### Nucleation bulges can confer binding affinity similar to that of canonical sites

In mammals, target insertions and deletions in the seed increase $K_D$[23,25,54]. An exception is nucleation-bulge sites identified by high-throughput crosslinking and immunoprecipitation in mouse brain[55]. These 8mer sites consist of seven nucleotides paired to miRNA positions g2–g8 and one nucleotide opposing position g6 protruding as a bulge but sharing potential complementarity to miRNA position g6.

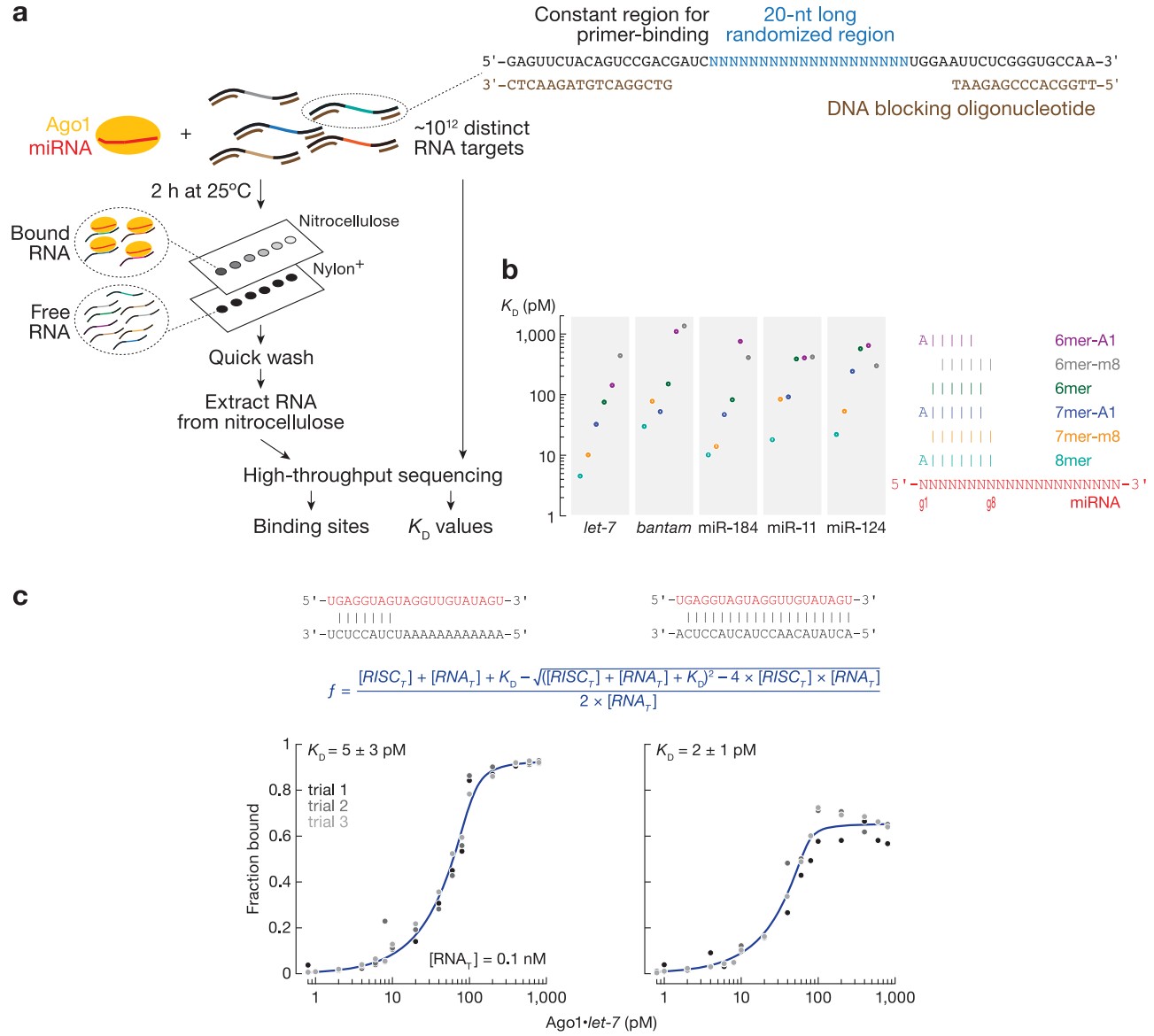

**Fig. 1 | RNA bind-n-seq (RBNS) measures binding affinities of canonical target sites. a** Overview of RBNS. **b** $K_D$ values fitted for canonical target sites. Error bars indicate 95% CI of the median from 2000 independent MLE runs. Right panel: pairing of target sites. **c** Equilibrium binding assays of Ago1•*let-7* for seed-matching and fully complementary target RNA using a double-filter binding assay. Data are mean ± SD for three independent experiments. Source data are provided as a Source data file.

**Table 1 | Similarities and differences between the targeting rules for miRNA-guided fly AGO1 and mammal AGO2**

| Feature | Fly miRNAs | Mammalian miRNAs | References |
|---|---|---|---|
| Seed pairing | Seed complementarity dominates DmAgo1 target binding | Canonical seed pairing is the most efficient way to reach high-affinity binding | This study[23–25,53] |
| G:U pairs in the seed | DmAgo1 does not tolerate G:U wobble pairs | AGO2 can display moderate affinity | This study[23,24,52] |
| 1-nt mismatches in the seed | Limited tolerance for mismatches, but some miRNAs may bind sites containing one single mismatch | Broader, the imperfections tending to occur at different positions, with affinities similar to those of the canonical sites | This study[23,24,52] |
| 1-nt bulges in the seed | Some nucleation-bulged sites are allowed | More diversity in bulged sites | This study[23,24,55] |
| 3' pairing | Required for stable interaction with mismatched seeds | Stabilizes pairing with weak or imperfect seeds | This study[23,35] |
| 3'-only sites | Some 3'-only sites bind with comparable or higher affinity than the canonical 6mer site | Some 3'-only sites bind with comparable or higher affinity than the canonical 6mer site | This study[24,25,74] |
| Centered sites | Can be cleaved by Ago1 | Can be cleaved by AGO2 | This study[26] |
| Identity of nucleotides flanking binding sites | Impacts binding affinity, likely by influencing site accessibility | Impacts $k_{on}$ and binding affinity, likely by influencing site accessibility | This study[23–25,62] |

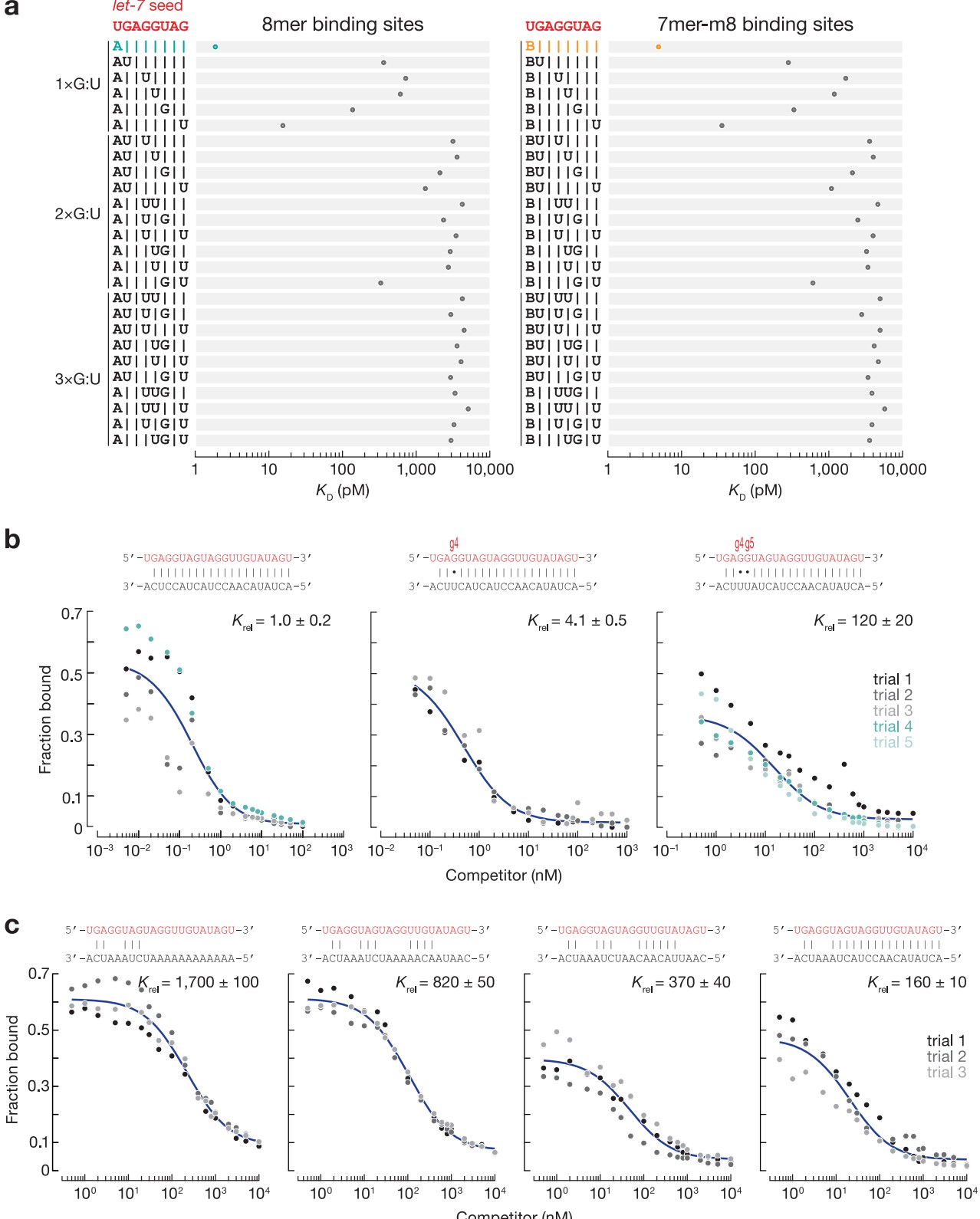

**Fig. 2 | The effect of G:U pairs within the seed on miRNA binding affinity. a** $K_D$ values for canonical seed-matched sites (left: t2–t8 targets with t1A; right: t2–t8 targets) and one, two and three G:U pairs within *let-7* seed at indicated positions. Shown for comparison: 8mer (t2–t8 site with t1A; in cyan) and 7mer-m8 (t2–t8 site; in orange). Error bars indicate 95% CI on the median from 2000 independent MLE runs. B represents C, G, or U. The equilibrium dissociation constant of *let-7*-loaded Ago1 for the competitor bearing G:U base-pairs (**b**) or mismatches (**c**) within the seed, relative to that of a fully complementary target. miRNA sequence shown in red. Data are mean ± SD for ≥three independent experiments. Source data are provided as a Source data file.

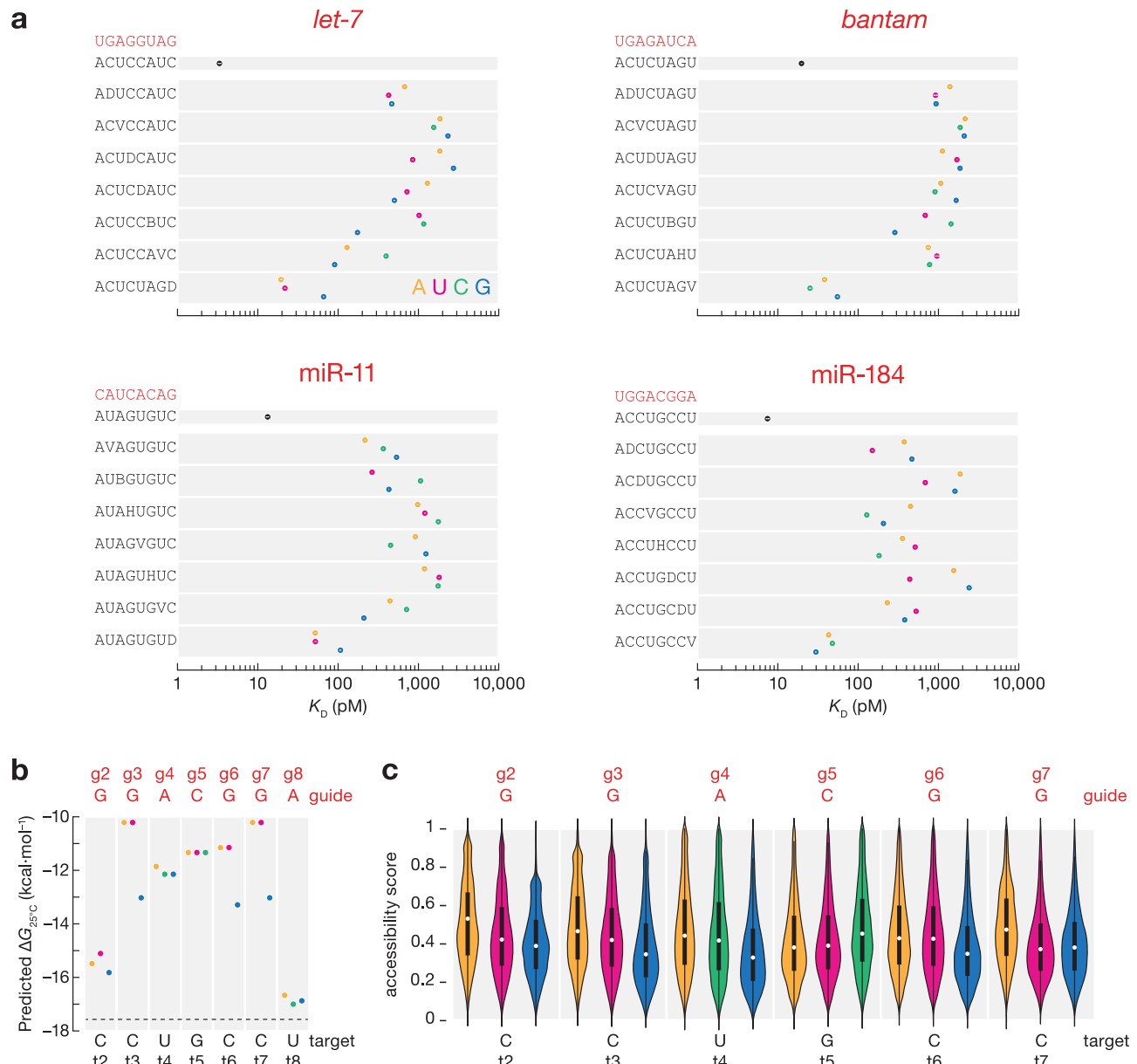

**Fig. 3 | The effect of target mismatches on miRNA binding affinity. a** $K_D$ values for t2–t8 targets with t1A and different one-nucleotide mismatches at indicated positions. Error bars indicate 95% CI of the median from 2000 independent MLE runs. B represents C, G, or U; D represents A, G, or U; H represents A, C, or U; V represents A, C, or G. Nearest neighbor free energy values (**b**) and accessibility scores (**c**) for miR-184 t2–t8 targets with t1A and different one-nucleotide mismatches at indicated positions. Violin plots show the kernel-density distributions of the data (>6800 data points per group). The embedded box indicates the interquartile range (25th–75th percentiles), with the central white dot marking the median. Adenine (orange), uridine (magenta), cytidine (green), and guanosine (blue). miRNA g1–g8 sequence shown in red. Horizontal dashed line in (**b**) indicates the nearest neighbor free energy for t2–t8 targets with t1A.

Some mammalian miRNAs effectively bind nucleation-bulge sites in vitro[24,25] and repress nucleation-bulge sites in cultured cells[55].

Like mammalian AGO2, fly Ago1 does not tolerate indels within the seed: small target deletions or 1-nt bulges reduced binding affinity by ~50–1000-fold (Figs. 4 and S3b, c). Nevertheless, nucleation-bulge sites displayed binding affinities similar to those of canonical sites: $K_{D,\text{8mer-b5.6A}}^{let-7} = 120$ pM vs. $K_{D,\text{6mer-A1}}^{let-7} = 92$ pM, $K_{D,\text{8mer-b5.6G}}^{miR-11} = 107$ pM vs. $K_{D,\text{7mer-A1}}^{miR-11} = 70$ pM, $K_{D,\text{8mer-b5.6C}}^{miR-184} = 38$ pM vs. $K_{D,\text{7mer-A1}}^{miR-184} = 37$ pM, and $K_{D,\text{8mer-b5.6G}}^{miR-124} = 75$ pM vs. $K_{D,\text{7mer-A1}}^{miR-124} = 242$ pM (Figs. 4b and S3c). Moreover, the nucleation-bulge sites across the five miRNAs tested bind at greater affinity than the corresponding 6mer-A1 sites with similar flanking nucleotide composition (Fig. S3d). Like in mammals, nucleation bulges may

constitute an alternative mode of *Drosophila* miRNA target recognition that follows a pivot-pairing rule.

### Repertoire of fly Ago1 target sites
In an RBNS experiment, Ago1 is incubated with a pool of $10^{12}$ distinct RNA sequences. Therefore, in addition to measuring binding affinities of specific sites of interest, RBNS enables a de novo search for sites of productive binding[24,25,36,56]. As expected, this approach identified canonical 8mer, 7mer-m8, 7mer-A1, 6mer, and 6mer-m8 binding sites (Fig. 5a). For *let-7*, it also identified short seed-matched sites 6mer-A1, 5mer-A1, and 5mer-m2.6, as well as 8mer-w8 and 7mer-w8–sites bearing a G:U wobble at the distal t8 position. Because fly Ago1 does not tolerate G:U pairing in the seed, these 8mer-w8 and 7mer-w8 likely

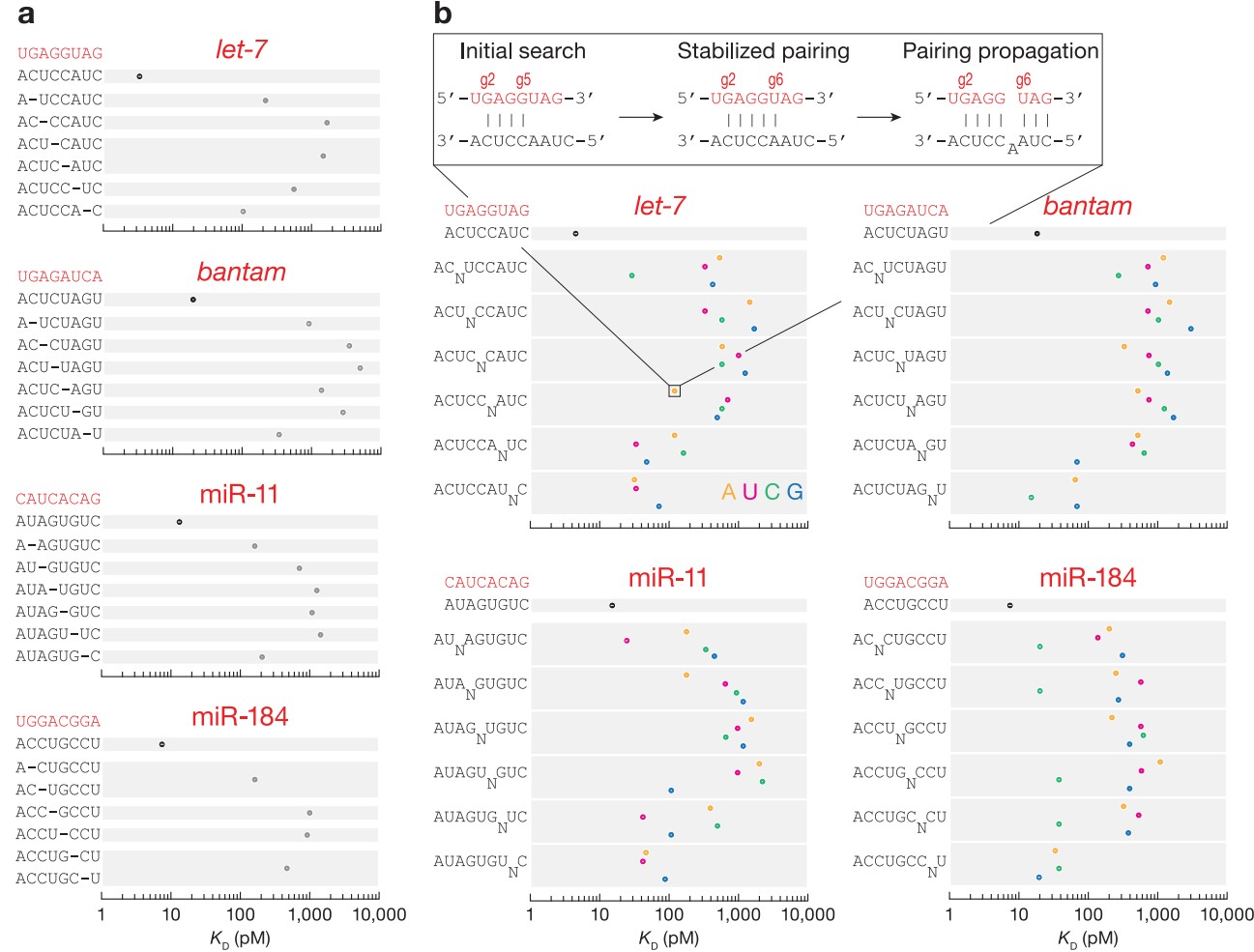

**Fig. 4 | $K_D$ values for t2–t8 targets with t1A and one-nucleotide deletions or bulges.** One-nucleotide deletions (**a**) and bulges (**b**) at indicated positions. Error bars indicate 95% CI of the median from 2000 independent MLE runs.

Adenine (orange), uridine (magenta), cytidine (green), and guanosine (blue). miRNA g1–g8 sequence shown in red.

correspond to 7mer-A1 and 6mer sites with favorable, AU-rich flanking contexts (see below). Motifs complementarity to positions g13–g16 were also detected. These likely correspond to 3' supplemental pairing for weak seeds that are too variable to be detected by our analyses: e.g., 5-mer sites separated from the supplemental pairing by loops of various sizes. The repertoire of binding sites for miR-11 resembled that of *let-7*, except that two additional short seed-matched sites, 6mer-m4.9 and 5mer-m4.8, could be identified. Sites with 3' supplemental pairing were not enriched above background in miR-124, miR-184 and *bantam* datasets. miR-184 sites included canonical, short seed-matched and 8mer-x4C–an 8mer site bearing a mismatched t4 nucleotide with $K_D$ ∼ 130 pM (Fig. 3a). Overrepresented sites bound by *bantam* included only canonical and 5mer-m2.6 sites. Our de novo site discovery algorithm identified only three binding sites for fly miR-124–loaded Ago1: the canonical 8mer, 7mer-m8, and 7mer-A1 (Fig. 5a). Remarkably, AGO2 guided by human miR-124, whose sequence is identical to that in flies, binds ∼15 site types within a 10-fold range of the 8mer affinity, spanning canonical and noncanonical interactions, including seed-matched sites with wobble pairs, mismatches, bulges, and 3'-only sites[24].

Considering these differences in site recognition, we examined the ΔG penalties associated with individual seed-region imperfections across the five fly miRNAs examined in this study and the mammalian miRNAs from previously published AGO2 RBNS datasets[24,25]. The binding affinity for a site is predicted to be more stable at 25 °C than

37 °C. Contrary to expectation, for miR-124 and let-7, which are shared between the fly and mammalian datasets, binding to a perfectly matched g2–g8 target was more stable for mammalian AGO2 at 37 °C than for fly Ago1 at 25 °C (Fig. S4). This observation suggests that the reduced tolerance for mismatches of fly Ago1 does not arise because mismatches are inherently more destabilizing. Instead, the overall dynamic range of binding starts with weaker binding. We also note that our binding experiments were performed using 3.5 mM Mg$^{2+}$. As expected, when the experiments were performed using 0.89 mM Mg$^{2+}$, the concentration used previously to study binding of human AGO2 (ref. 24), we recovered fewer site-types (Fig. S5a). Together, our data suggest that fly Ago1 has lower tolerance for pairing imperfection because it binds perfect seed matches less tightly than AGO2.

**Fly Ago1 cleaves central sites and binds some 3'-only sites**

Unlike high-throughput analysis of mammalian AGO2[24,25], our RBNS experiments with fly Ago1 failed to detect central or 3'-only sites. Mammalian AGO2 can cleave targets with centered sites, sites with 11–12 nucleotides complementarity to the center of the miRNA[26]. *Drosophila* Ago1 retains the ability to catalyze endonucleolytic cleavage of extensively matched target RNAs, and like other Argonaute proteins, cleaves the phosphodiester bond linking target nucleotides t10 and t11[57–59]. In theory, efficient cleavage of targets with centered sites might prevent their recovery in RBNS data. To test this idea, we performed in vitro cleavage assays with synthetic RNA targets

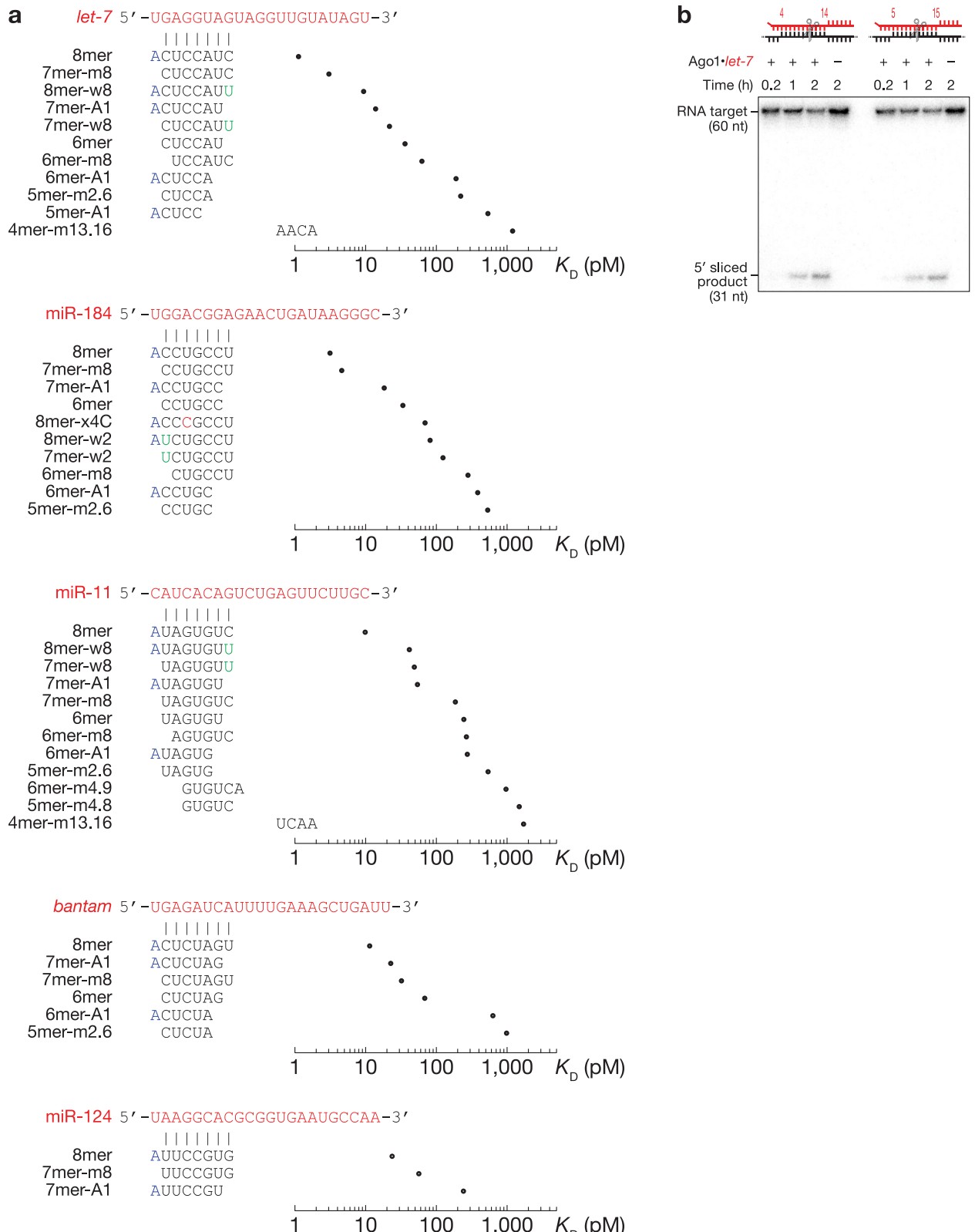

**Fig. 5 | Sequence-specific differences in miRNA affinities. a** Site types identified by de novo site discovery in Ago1 *let-7*, miR-184, miR-11, *bantam*, and miR-124 RBNS and their fitted dissociation constants. Left panel: pairing of each site, indicating Watson–Crick pairing (black), wobble pairing (green), mismatched pairing (red), and t1A (blue). **b** Representative denaturing polyacrylamide gel electrophoresis showing cleavage of centered sites 11mer-m4.15 and 11mer-m5.15 by *let-7*-loaded Ago1. See Supplementary Fig. S5c. Three independent experiments. Source data are provided as a Source data file.

containing two distinct 11-nt centered sites complementary to miRNA nucleotides g4–g14 (11mer-m4.14) or g5–g15 (11mer-m5.15). Ago1•*let-7* cleaved both centered sites at levels comparable to those observed with a fully complementary RNA (Figs. 5b and S5c, d). Therefore, cleavage of central sites likely prevents their recovery in RBNS libraries, making RBNS unsuitable for measuring their binding affinities. We note that mammalian AGO2-catalyzed mRNA cleavage of 11–12-nt-long centered sites requires $Mg^{2+}$ >2 mM[26]; in animals, the intracellular $Mg^{2+}$ concentration is <1 mM[60].

By contrast, some 3′-only sites were enriched in RBNS datasets but did not pass our criteria for significance (see "Materials and Methods"). Therefore, we directly measured the affinity of sites with 10-nt segments of perfect complementarity to the miRNA sequence, scanning from position g9 to the 3′ end of the miRNA (Fig. S5b). For *bantam*, miR-184, miR-12,4 and miR-11, 10-mers that base paired to the guide 3′ end bound with comparable or higher affinity than the canonical 6mer site. Except for the *bantam* 10mer-m14.23 site, motifs with more distal pairing had higher affinities than more seed-proximal sites: $K_{D,10mer-m9.18} = 587$ pM, $K_{D,10mer-m10.19} = 752$ pM, $K_{D,10mer-m11.20} = 571$ pM, $K_{D,10mer-m12.21} = 217$ pM, and $K_{D,10mer-m13.22} = 71$ pM vs. $K_{D,6mer} = 112$ pM. For *let-7*, none of the 10-mers conferred stronger binding than a canonical 6mer.

### Influence of flanking dinucleotide sequences

The nucleotide composition of sequences immediately flanking miRNA sites significantly influences mRNA repression in mammalian cells[18,21,23,24,61]. Local sequence context similarly influences Ago1 binding affinity. We divided the *let-7* 8mer site into $4^4 = 256$ different 12-nt sites according to the dinucleotide sequences immediately flanking the 5′ and the 3′ ends of the 8mer site and measured their $K_D$ values by RBNS. We used RNAplfold[62] to estimate the accessibility of each site by computing the probability of its being unpaired at thermodynamic equilibrium. The predicted accessibility scores were negatively correlated to measured binding affinities (Pearson's $r = -0.352$, $p$-value < 0.001; Fig. 6). For the 256 distinct flanking contexts, the $K_D$ values spanned a ~ 20-fold range, with higher binding affinities corresponding to greater AU content in the flanking dinucleotides.

We repeated our analysis with 8mer binding sites for *bantam*, miR-184, and miR-11. As we observed for *let-7*, 8mer binding sites with low accessibility scores had low binding affinities for all three miRNAs (Fig. 6). Finally, for every miRNA tested, we divided 7mer-m8, 7mer-A1, 6mer, 6mer-m8, and 6mer-A1 binding sites into 256 different sites according to the dinucleotide sequences immediately flanking the 5′ and the 3′ ends of each site type and measured their $K_D$ values by RBNS. The binding affinities for nearly all site types correlated with accessibility scores. The exceptions were *let-7*, *bantam*, miR-11, miR-124 binding to 6mer-A1 sites, miR-11 and miR-124 binding to 7mer-A1 sites, and miR-124 binding to 6mer sites. These data suggest that *Drosophila* Ago1, like mammalian AGO2[23,24], competes with local RNA secondary structures for site binding.

## Discussion

Despite the importance of *D. melanogaster* as a model system, the targets of most fly miRNAs are largely unknown. The most successful prediction algorithm, TargetScan Fly v7 explains only a small fraction of changes in mRNA abundance caused by the introduction of a miRNA to S2 cells (coefficient of variation $r^2 = 0.19$)[27]. Recent improvement of models predicting miRNA-mediated silencing in mammalian cells required quantitative, high-throughput measurements of miRNA binding affinities[23,24]. To date, most studies of fly miRNAs have used reporter assays in flies and examined repression efficacy of individual targets one-by-one[15,16,29]. RNA sequencing using *Drosophila* S2 cells monitored changes in steady-state levels of all cellular mRNAs after cell transfection with supraphysiological concentrations of exogenous miRNA duplexes[27]. Higher-than-physiological miRNA concentrations

may identify binding sites that are not functional in vivo. High-throughput sequencing methods that rely on UV crosslinking and immunoprecipitation yield comprehensive lists of RNA-binding motifs but do not enable quantitative assessment of binding affinities[63]. Consequently, biochemically validated rules for predicting the targets of miRNAs in flies still do not exist.

The first step in constructing a predictive model requires quantitative measurements of binding affinities between miRNA-loaded Ago proteins and target sites. Higher affinity binding is expected to yield greater occupancy of a target site and therefore greater repression. We used RBNS to identify productive binding sites for five fly miRNAs. Our analyses using stringent statistical thresholds revealed that the repertoire of fly miRNA target sites is less complex than in mammals. Fly miRNAs bind canonical and seed-matched binding sites and poorly tolerate imperfections within the seed. Nevertheless, seed-matched sites with a single imperfection (mismatch, wobble pairing, 1-nt target deletion or 1-nt target insertion) could be compensated by extensive pairing to the guide sequence, as suggested by reporter assays in flies[16]. Consistent with this idea, motifs complementary to guide nucleotides at g13–g16 were enriched in *let-7* and miR-11 RBNS datasets. However, the maximum effective motif size for de novo site discovery by RBNS is 10 nt, because reads with longer motifs are sparse in the sequencing datasets. Therefore, identifying the minimal degree of complementarity required to rescue weak canonical sites or seed-matched sites with imperfections in flies will call for other experimental strategies such as RBNS modified to interrogate bipartite sites[35] or screen large numbers of pre-designed RNA sequences using TIRF[23]. Structural and biochemical studies of mammalian miRNAs revealed that the seed and 3′ supplementary regions can be bridged by target loops 7–15-nt long[23,54]. If fly Ago1 provides a similar mechanism for extended miRNA targeting, compensatory sites in flies may include a larger number of transcripts than previously estimated.

While fly miRNA binding is more restricted than in mammals, fly miRNAs bound some non-canonical seed-matched sites with similar or greater affinity than a canonical 6mer site. One of these was the nucleation bulge site[55], which was bound by *let-7*, miR-11, miR-184, and miR-124, but not by *bantam*. Another example was a miR-184 binding site with extended seed and a cytosine at position t4 (8mer-x4C). This site type was productively bound only by miR-184; whether other miRNAs can bind seed-matched sites with single nucleotide mismatches remains to be uncovered. Finally, our data suggest that miRNAs may effectively target seedless 3′-only and central non-canonical sites, as observed for mammalian miRNAs[24–26].

The ability to measure binding affinities from RBNS datasets enabled us to obtain $K_D$ values for hundreds miRNA binding sites. Canonical binding sites of the five miRNAs tested displayed miRNA-specific differences in their binding affinities. We envision that our high-throughput measurements of binding affinities using RBNS may enable prediction of changes in the occupancy of fly miRNA-guided Ago1 at sites across the transcriptome in response to developmental stimuli, thereby allowing the modeling of the resulting changes in regulatory activity.

## Methods

### RNA

Ago1 was loaded with chemically synthesized (Sigma) guide and the passenger strands list in Table S1; each synthetic RNA was gel-purified before use. The siRNA duplexes were annealed for 1 min at 90 °C, 1 h at 37 °C and 10 min at room temperature (20–25 °C) in 30 mM HEPES-KOH, pH 7.9, 100 mM potassium acetate, 2 mM DTT using a 1:1.25 molar ratio of guide-to-passenger.

Libraries of RNA oligonucleotides, each containing 20 central random nucleotides (Table S1), were synthesized (IDT) with an equal ratio of bases for each random position, 5′ $^{32}$P-labeled and gel-purified. DNA "blocker" oligonucleotides complementary to the common

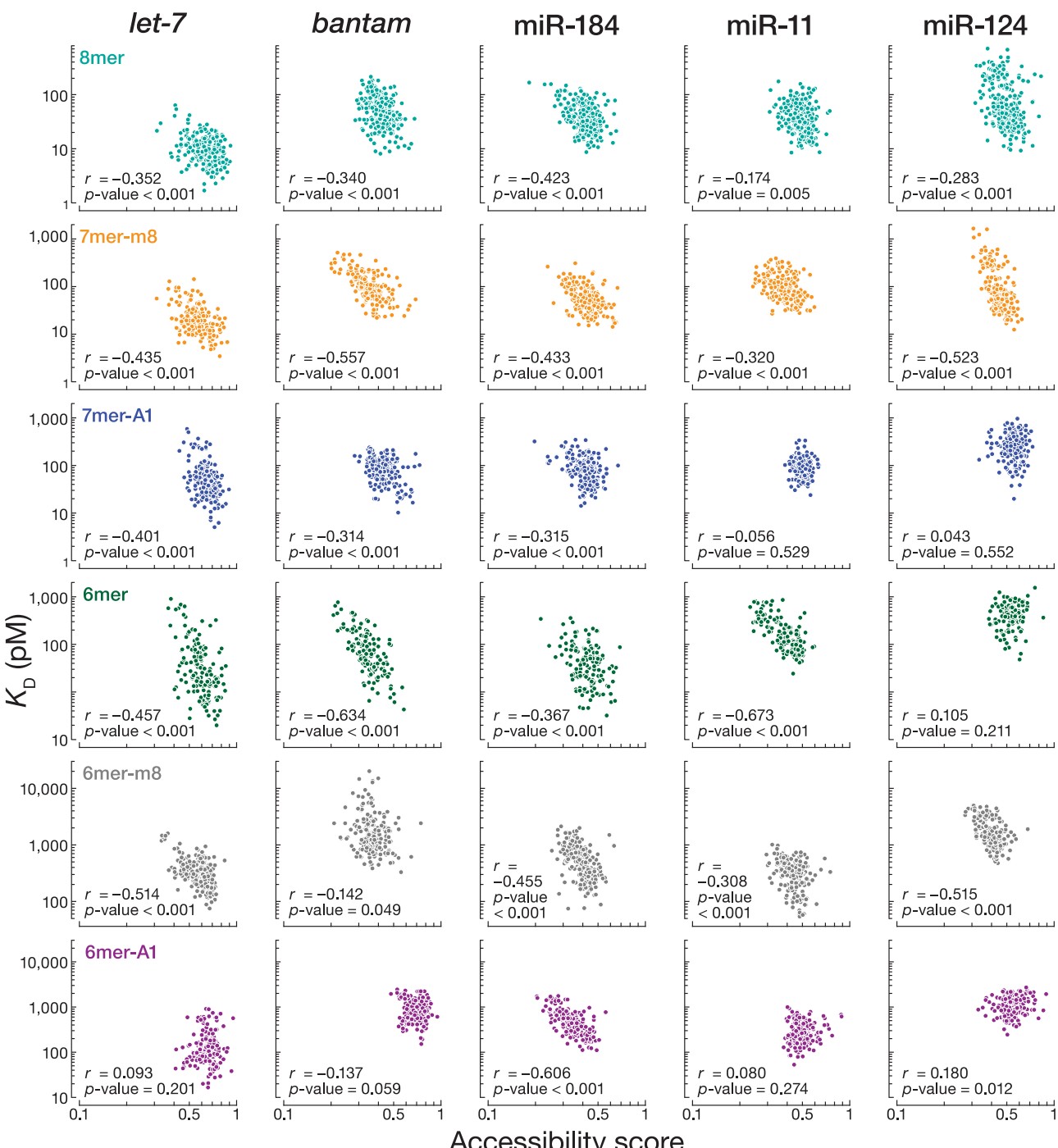

**Fig. 6 | The influence of flanking dinucleotide sequences.** Relationship between the accessibility score and $K_D$ values for the 256 sites containing one canonical target site flanked by each of the dinucleotide combinations. Our analysis included 8mer, 7mer-m8, 7mer-A1, 6mer, 6mer-m8, and 6mer-A1 for all five miRNAs. $r$ and $p$-values were calculated using Pearson correlation (two-sided test). Exact $p$-values are provided as a Source data file.

sequences present in each RNA molecule were synthesized (IDT) and annealed to the RNA library in 30 mM HEPES-KOH, pH 7.5, 120 mM potassium acetate, 3.5 mM magnesium acetate using a 1:1.2 molar ratio of RNA pool to DNA blockers; 5′-end and 3′-end blocking cDNA oligonucleotides #1 (Table S1) were used in RBNS for Ago1 loaded with *let-7*, *bantam*; 5′-end and 3′-end blocking cDNA oligonucleotides #2 were used in for Ago1 loaded with miR-184 (Table S1).

## Cell culture
*Drosophila* Schneider 2 (S2) cells (Gibco, Catalog No. R690-07) were cultured in *Drosophila* Schneider's media (Gibco) supplemented with

10% FBS. Cells were maintained in a humidified incubator at 27 °C and examined regularly to ensure absence of mycoplasma contamination.

## Generation of FLAG-Ago1 expressing cells
Ago1 cDNA was amplified by RT-PCR from total RNA extracted from $w^{1118}$ fly heads ($w^{1118}$ flies were obtained from the KYOTO *Drosophila* Stock Center, Kyoto Institute of Technology). The Ago1 coding sequence was cloned into pMT expression vector (Invitrogen), fusing Ago1 in-frame with an N-terminal FLAG tag. The resulting plasmid is available from Addgene (#229236). S2 cells (25 million, 70% confluent) were transfected with 10 μg pMT-FLAG-Ago1 plasmid using 20 μl

TransIT-Insect Transfection Reagent (Mirus Bio) according to manufacturer's instructions. After 18 h transfection, copper sulfate was directly added to the culture medium to a final concentration of 500 μM, and the cells were incubated for 30 h.

## RISC purification

FLAG-Ago1 expressing S2 cells were harvested; cell pellets were flash-frozen and stored at −80 °C. Cell extract was essentially prepared as described[64]. Briefly, the cell pellet was washed three times in ice-cold PBS and once in Buffer A (10 mM HEPES-KOH, pH 7.9, 10 mM potassium acetate, 1.5 mM magnesium acetate, 0.01% (w/v) CHAPS, 0.5 mM DTT, 1 mM AEBSF, hydrochloride, 0.3 μM Aprotinin, 40 μM Bestatin, hydrochloride, 10 μM E-64, 10 μM Leupeptin hemisulfate). Then, the pellet was resuspended in twice its volume with buffer A and incubated on ice for 20 min to allow the cells to swell. The cells were subsequently lysed with a Dounce homogenizer and a tight pestle (B type; 40 strokes) on ice. The homogenate was centrifuged at 2000 × *g* to remove nuclei and cell membranes. Next, 0.11 volumes (that of the clarified supernatant from the low-speed centrifugation) of Buffer B (300 mM HEPES-KOH, pH 7.9, 1.4 M potassium acetate, 30 mM magnesium acetate, 0.01% (w/v) CHAPS, 0.5 mM DTT, 1 mM AEBSF, hydrochloride, 0.3 μM Aprotinin, 40 μM Bestatin, hydrochloride, 10 μM E-64, 10 μM Leupeptin, hemisulfate) was added, followed by centrifugation at 100,000 × *g* for 20 min at 4 °C and the supernatant collected as the S100 extract. Finally, 0.32 volumes ice-cold Buffer C (30 mM HEPES-KOH pH 7.9, 120 mM potassium acetate, 3.5 mM magnesium acetate, 0.01% (w/v) CHAPS, 2 mM DTT, 1 mM AEBSF, hydrochloride, 0.3 μM Aprotinin, 40 μM Bestatin, hydrochloride, 10 μM E-64, 10 μM Leupeptin, hemisulfate) was added to the S100 to achieve a 20% (w/v) final glycerol concentration, followed by gentle inversion to mix. The S100 was aliquoted, frozen in liquid nitrogen, and stored at –80 °C.

To assemble RISC, 3 ml S100 (~40 mg total protein) was incubated with 150 nM miRNA duplex for 1.5 h at 25 °C in 30 mM HEPES-KOH pH 7.9, 120 mM potassium acetate, 3 mM magnesium acetate, 2 mM DTT, 1 mM ATP, 25 mM creatine phosphate, 30 μg/ml creatine kinase.

To capture RISC, 30 μl anti-FLAG M2 paramagnetic beads (Sigma) were used per ml of post-assembled extract, washed three times with 30 mM HEPES-KOH pH 7.9, 120 mM potassium acetate, 3 mM magnesium acetate, 0.05% (w/v) CHAPS, 2 mM DTT and incubated with assembled RISC with rotation at 4 °C overnight. Beads were washed five times with wash buffer (30 mM HEPES-KOH, pH 7.9, 120 mM potassium acetate, 3 mM magnesium acetate, 2 mM DTT, 0.05% (w/v) CHAPS), and RISC was eluted from the beads with 100 ng/ml 3XFLAG peptide (Sigma) in 300 μl wash buffer for 1 h at room temperature. Elution was repeated two more times, and the three eluates combined. RISC was further purified via the by sequence affinity capture[65]. Briefly, RISC was incubated with gentle rotation for 1 h at room temperature with 800 pM biotinylated, 2′-O-methyl capture oligonucleotide (Table S1) linked to streptavidin paramagnetic beads (Dynabeads MyOne Streptavidin T1, Life Technologies). RISC was eluted with 2 nmol biotinylated competitor oligonucleotide (Table S1) in 80 μl buffer composed of 30 mM HEPES-KOH pH 7.9, 1 M potassium acetate, 3.5 mM magnesium acetate, 0.05% (w/v) CHAPS, 20% (w/v) glycerol, 2 mM DTT for 2 h at room temperature. Excess competitor oligonucleotide was removed by incubating the eluate with 3.5 mg streptavidin paramagnetic beads (Dynabeads MyOne Streptavidin T1, Life Technologies) for 15 min at room temperature. Finally, RISC was dialyzed at 4 °C against three changes (3 h each) of a 3000-fold excess of storage buffer (30 mM HEPES-KOH pH 7.9, 120 mM potassium acetate, 3.5 mM magnesium acetate, 0.05% (w/v) CHAPS, 2 mM DTT, 20% (w/v) glycerol). RISC was aliquoted, frozen in liquid nitrogen, and stored at –80 °C. For competition experiments (Fig. 2b, c), RISC was assembled in *Drosophila* embryo lysate [w[*]; AGO2[414]]; Stock Number (109027);

Collection (Kyoto Stock Center)] from 0 to 8-h embryos prepared as described[66]; assembled RISC was purified as described above[65].

## Quantification of purified RISC

Total concentration of RISC was estimated by measuring concentration of associated miRNA using Northern blotting as described[67]. Briefly, miRNA guide standards and RISC were first resolved on a denaturing 15% polyacrylamide gel, followed by transfer to Hybond-XL (Cytiva) nylon membrane by semi-dry transfer at 20 V for 1 h. Next, crosslinking was performed with 0.16 M 1-ethyl-3-(3-dimethylamino-propyl) carbodiimide in 0.13 M 1-methylimidazole, pH 8.0, at 60 °C for 1 h. The crosslinked membrane was pre-hybridized in Church buffer (1% (w/v) UltraPure BSA, 1 mM EDTA, 0.5 M phosphate buffer, and 7% (w/v) SDS) at 37 °C for 1 h. Next, 25 pmol 5′ $^{32}$P-radiolabeled RNA probe (Table S1) in Church buffer was added to the membrane and allowed to hybridize overnight at 37 °C, followed by two washes with 1× SSC containing 0.1% (w/v) SDS for 5 min and two washes with 0.1× SSC containing 0.1% (w/v) SDS at 37 °C for 15 min. The membrane was air dried and exposed to a storage phosphor screen.

The concentration of active RISC was measured by stoichiometric binding titration assays as described[51]. Binding reactions were performed in 5 μl in the presence of 30 mM HEPES-KOH, pH 7.9, 120 mM potassium acetate, 3.5 mM magnesium acetate, 2 mM DTT, 0.01% (w/v) CHAPS. Concentration of 5′ $^{32}$P-RNA target complementary to the seed region of the miRNA guide (Table S1) was 0.5 nM in all reactions, and the RISC concentrations spanned from 0.05 to 5 nM. The assay also included a mock binding reaction using RISC storage buffer. Binding reactions were incubated at 25 °C for 1 h. RNA binding was measured by capturing protein-RNA complexes on Protran nitrocellulose membrane (Whatman, GE Healthcare Bioscience, Pittsburgh, PA) and unbound RNA on a Hybond-XL (Cytiva) nylon membrane with a Bio-Dot apparatus (Bio-Rad, Hercules, CA). After applying the sample under vacuum, membranes were washed with 10 μl of equilibration buffer (30 mM HEPES-KOH pH 7.9, 120 mM potassium acetate, 3.5 mM magnesium acetate, 2 mM DTT). Membranes were air-dried and signals detected by phosphorimaging. To measure concentration of active, binding-competent RISC, titration data were fit to

$$f(r) = f_{max} \times \frac{r + K_D + n - \sqrt{(r + K_D + n)^2 - 4 \times r \times n}}{2 \times n}$$

where $K_D$ is the apparent dissociation constant, $r$ is the molar ratio of [RISC] to [RNA], $n$ is the stoichiometric equivalence point, $f$ is the fraction bound, $f_{max}$ is the maximum fraction bound[68].

## Competition assays

Competition reactions (5 μl) were performed in 18 mM HEPES-KOH, pH 7.4, 100 mM potassium acetate, 3 mM magnesium acetate, 5 mM DTT, 0.01% (v/v) IGEPAL CA-630, 0.01 mg/ml baker's yeast tRNA. Both 5′ $^{32}$P-RNA target fully complementary to the miRNA guide (Table S1) and RISC were 0.1 nM in all reactions. Concentrations of the competitor RNA (Table S1) were spanned from 0.02 nM–100 nM (target to *let-7* with g2–21 complementarity), 0.05–1000 nM (target to *let-7* with g2–21 complementarity and a G:U pair at position 4), and 0.5–10,000 nM (all the other *let-7* targets). The assay also included a no-RISC competition reaction using binding buffer. All reactions were incubated at 25 °C for 1 h. The fraction bound was measured by capturing protein-RNA complexes on Protran nitrocellulose membrane (Whatman, GE Healthcare Bioscience, Pittsburgh, PA) and unbound RNA on a Nylon XL membrane (GE Healthcare Bioscience) using a Bio-Dot apparatus (Bio-Rad, Hercules, CA). The sample was placed on the top of the membranes and after applying vacuum, the membranes were washed with 10 μl of ice-cold wash buffer (18 mM HEPES-KOH, pH 7.9, 100 mM potassium acetate, 3 mM magnesium acetate, 5 mM DTT, 0.01 mg/ml baker's yeast tRNA). Membranes were air-dried and signals

detected by phosphorimaging. Data were fit to

$$\theta = \frac{\left([E_T] + [S_T] + K_D + \frac{K_D \times [C_T]}{K_C}\right) - \sqrt{\left([E_T] + [S_T] + K_D + \frac{K_D \times [C_T]}{K_C}\right)^2 - 4[E_T] \times [S_T]}}{2 \times [S_T]}$$

where $[E_T]$ is total enzyme concentration, $[S_T]$ is total RNA target concentration, $K_D$ is the apparent equilibrium dissociation constant, $\theta$ is the fraction target bound in presence of competitor RNA with an apparent dissociation constant of $K_C$, $[C_T]$ is total competitor concentration[69,70].

### Target cleavage assays

Purified Ago1 miRISC (0.32 nM final concentration) was incubated at 25 °C with 5′-$^{32}$P-radiolabelled target RNA. The reactions employed 5 nM (Figs. 5b and S5c) or 100 nM (Fig. S5d) target RNA in reaction buffer composed of 25 mM HEPES-KOH, pH 7.5, 110 mM potassium acetate, 3.5 mM magnesium acetate, 2 mM DTT, 0.02% (w/v) CHAPS, 0.01 mg/ml baker's yeast tRNA, 1 U/µl RNasin, 8% (w/v) glycerol. At the indicated times, an aliquot of a master reaction was quenched in four volumes 100 mM Tris-HCl, pH 7.5, 200 mM NaCl, 20 mM EDTA, 2% (w/v) sodium dodecyl sulfate. Then proteinase K (1 mg/ml final concentration) was added and incubated at 55 °C for 20 min. An equal volume of formamide loading buffer (98% (w/v) formamide, 0.025% (w/v) xylene cyanol, 0.025% (w/v) bromophenol blue, 25 mM EDTA, pH 8.0) was added. Samples were incubated at 95 °C for 3 min and analyzed by electrophoresis through a denaturing 15% polyacrylamide 7 M urea gel using 0.5× Tris-borate-EDTA buffer. Gels were dried, exposed to a storage phosphor screen, and imaged on a Typhoon FLA7000IR phosphorimager (GEHealthcare).

### Measuring equilibrium dissociation constants by double filter-binding assays

Binding assays were performed as previously described[51] in 5 µl in equilibration buffer (25 mM HEPES-KOH, pH 7.9, 110 mM potassium acetate, 3.5 mM magnesium acetate, 2 mM DTT, 0.05 µg/µl BSA, 0.01 mg/ml baker's yeast tRNA, 8% (w/v) glycerol, 1 U/µl RNasin Plus). The 5′ [$^{32}$P]-radiolabeled RNA targets (0.1 nM; Table S1) were incubated with a RISC concentration between 0.0008 and 0.8 nM RISC. The assay also included a control reaction using equilibration buffer. Binding reactions were incubated at 25 °C for 2 h. RNA binding was measured by capturing protein–RNA complexes on Protran nitrocellulose (GE, GE10600002) and unbound RNA on a Hybond-XL (Cytiva, 45001151) in a Bio-Dot apparatus (Bio-Rad) as previously described[71]. After applying the sample under vacuum, membranes were washed with 10 µl of wash buffer (25 mM HEPES-KOH, pH 7.5, 110 mM potassium acetate, 3.5 mM magnesium acetate and 2 mM DTT). Membranes were air-dried and signals detected by phosphorimaging.

Because $K_D <$ [RNA target], all binding data were fit to the following equation using IgorPro9.05 (WaveMetrics):

$$f = \frac{\left([E_T] + [S_T] + K_D\right) - \sqrt{\left([E_T] + [S_T] + K_D\right)^2 - 4[E_T][S_T]}}{2[S_T]}$$

where $f$ is the fraction target bound, $[E_T]$ is the total RISC concentration, $[S_T]$ is the total RNA target concentration and $K_D$ is the apparent equilibrium dissociation constant.

### RNA bind-n-seq for de novo site discovery and $K_D$ measurements

RBNS were essentially performed as described[25,71].

Each experiment included five or six binding reactions. The highest concentration of RISC used corresponded to 40% (v/v) of the stock solution and equaled 4.0–5.8 nM (f.c.) active protein. For additional reactions, the stock was serially diluted 3.2-fold in storage

buffer. Each experiment also included a mock binding reaction (no-RISC control) using protein storage buffer without RISC. For each miRNA, we performed an additional binding reaction using protein storage buffer with miRNA guide at the highest miRISC concentration assayed, but lacking Ago1 protein. All binding reactions (20 µl) were performed in 20 mM HEPES-KOH, pH 7.9, 100 mM potassium acetate, 3.5 mM magnesium acetate, 0.05% (w/v) CHAPS, 2 mM DTT, 8% (w/v) glycerol, and contained 100 nM (f.c.) RNA library. To reduce non-specific binding, each reaction also included 0.05 µg/µl BSA and 0.01 mg/ml baker's yeast tRNA. Reactions were incubated for 2 h at 25 °C and then filtered through a Protran nitrocellulose membrane (Whatman, GE Healthcare Bioscience, Pittsburgh, PA) on top of a Hybond-XL (Cytiva) nylon membrane in a Bio-Dot apparatus (Bio-Rad, Hercules, CA). To reduce retention of free single-stranded RNA, we pre-conditioned nitrocellulose and nylon membranes prior to use as described[72,73]. Nitrocellulose filters were pre-soaked in 0.4 M potassium hydroxide for 10 min. Nylon filters were incubated in 0.1 M EDTA, pH 8.2 for 10 min, washed three times in 1 M sodium chloride for 10 min each followed by a quick rinse (-15 s) in 0.5 M sodium hydroxide. Nitrocellulose and nylon filters were then rinsed in water until the pH returned to neutral and equilibrated in wash buffer (20 mM HEPES-KOH, pH 7.9, 100 mM potassium acetate, 3.5 mM magnesium acetate, 2 mM DTT) for at least 1 h at 25 °C. After applying the sample under vacuum, membranes were washed with 100 µl wash buffer for 3 s. Membranes were air-dried and signals detected by phosphorimaging to monitor binding. The nitrocellulose membranes containing RISC-bound RNA were excised and incubated with 1 µg/µl Proteinase K (Thermo Fischer) in 100 mM Tris-HCl, pH 7.5, 10 mM EDTA, 150 mM sodium chloride, 1% (w/v) SDS, and 0.05 µg/µl glycogen for 1 h at 45 °C shaking at 300 rpm. After phenol-chloroform extraction and ethanol precipitation, RNA was denatured at 90 °C for 1 min, annealed to BRTP primer (Table S1) and reverse transcribed using SuperScript III (Life Technologies). RNA was degraded by alkaline hydrolysis using 0.4 M sodium hydroxide for 1 h at 55 °C, and cDNA was recovered by ethanol precipitation. The sample was then amplified with AccuPrime Pfx DNA Polymerase (Invitrogen). The reactions were run on a 2% agarose gel, amplicons were purified, then sequenced using a NextSeq 500 (Illumina) to obtain 75-nt, single-end reads.

Only Illumina reads containing TGG (the first nucleotides of the 3′ adapter) at positions 21–23 were analyzed. Sequences were filtered (Phred quality score ≥20 for all nucleotides, and "N" base calls disallowed), and the 3′ adapter sequence (5′ TGG AAT TCT CGG GTG CCA AGG 3′) removed.

Occurrences of all 10-nt long motifs (10-mers) were counted in all the reads of each RBNS sample. These counts were then divided by the total count of all 10-mers to give motif frequencies. Enrichment of a motif was computed as the ratio of the motif frequency in the protein-bound samples over the frequency in the RNA pool. Z-score of a motif was computed as $Z = \frac{R - \bar{R}}{S}$ where $R$ is enrichment of the motif, $\bar{R}$ is the mean of enrichment values of all 10-mers, and $S$ is the sample standard deviation of enrichment values of all 10-mers. A motif was considered significant if its Z-score was ≥99.9 percentile and was not enriched in the no-RISC control reaction.

Enrichments in the library from the binding reaction with the greatest RISC concentration were used for the following iterative procedure: (1) enrichment values of all 10-mers were calculated; (2) the hundred most enriched 10-mers were interrogated for base-pairing with the guide miRNA; (3) the most enriched site type was identified; (4) Z-scores of motifs belonging to the site type were compared to the Z-score threshold; (5) all reads containing the binding site were masked in the RISC-bound library and the RNA pool so that stepwise enrichments of subsequent 10-mers could be used to eliminate subsequent 'shadow' motifs; (6) all enrichment values were then recalculated on the masked read sets to obtain the resulting most enriched 10-mers. This process continued until the Z-score of the most enriched

binding site (calculated from the original enrichment values) was <99.9th percentile.

To identify a binding site at each iteration, the one hundred most enriched 10-mers were tested for base-pairing with the guide RNAs. If perfect complementarity was not observed, the 10-mer was tested for any of the following in this order: (1) complementarity to nine contiguous miRNA positions, allowing a single bulged target nucleotide; (2) complementarity to ten contiguous miRNA positions while allowing for wobble pairing; (3) complementarity to ten contiguous miRNA positions while allowing a non-wobble mismatch. If none of these configurations allowed assigning the motif to a binding site, the procedure was repeated with two 9-mers within the 10-mer, the three 8-mers within the 10-mer, etc., until a configuration of base-pairing was identified.

Each sequencing read in RNA pool and RISC-bound libraries was interrogated for presence of all binding sites of interest. The entire single-stranded sequence was interrogated: the 20-nt random-sequence region flanked by constant primer-binding sequences in the case when blockers were not used and the 20-nt random-sequence region flanked by 4 or 6 nucleotides of constant primer-binding sequence on either side in the case when blockers were annealed to the RNA pool. A read was assigned to a site category if it contained one single binding motif. Reads containing multiple instances of binding sites (from the same or a different site category) and reads containing partially overlapping sites were not included in the analysis and represented ≤1% of libraries. Reads that did not have any of binding motifs of interest were classified as reads with a no-site.

To estimate $K_D$ values, run the code[25] by following instructions in the README file. Bootstrapping of 95% of the data was performed ten times on sequencing reads from each binding reaction and the RNA pool. MLE of $K_D$ values was performed on each bootstrapped sample by using 100 different combinations of 10 initial guesses of miRISC concentration (in the range 0.5–25 nM) and 10 initial guesses of $K_D$ for RNA with no enriched site (in the range 0.5–10 nM). $K_D$ values were initialized as the inverse of the average enrichment values. The background was initialized at 0.1 nM. All the initial guesses were partially randomized by adding a value drawn from a normal distribution with mean 0 and standard deviation 0.1. The cost function was evaluated in the presence of physically meaningful constraints on the parameters: $0.1\,pM \leq K_D^{site} \leq 100\,nM$, $100\,pM \leq K_D^{no\text{-}site} \leq 10{,}000\,nM$, $100\,pM \leq miRISC \leq 100\,nM$, and $5\,pM \leq background \leq 5\,nM$. Any of the fitted parameters were at the boundaries at the end of the optimization routine. $K_D$ estimates, the background, and the stock concentration of miRISC provided by MLE were used to predict counts of each binding site type in sequencing data. These counts were compared with observed sequencing data, and MLE results were retained if Pearson correlation coefficient was >0.90. Results from independent starting points satisfying this criterion were combined. All bootstrapped samples were combined. Finally, estimates from two independent RBNS assays were merged. Median and 95% confidence intervals on medians were reported.

To estimate $K_D$ values of PUM2, fastq files of ENCFF761JAF, ENCFF894MLG, ENCFF090QVU, ENCFF574TBN, ENCFF266PHU, and ENCFF345UJY datasets were downloaded from the ENCODE project. Reads containing one single PUM binding site—UGUAUAUA (consensus binding site), AGUAUAUA, CGUAUAUA, and GGUAUAUA—were identified. Reads containing multiple instances of the PUM binding sites were not included in the analysis and represented ≤0.001% of libraries. Reads that did not have any of the above binding motifs were classified as reads with a no-site. MLE of PUM2 $K_D$ values was performed as described for miRISC.

## Reporting Summary

Further information on research design is available in the Nature Portfolio Reporting Summary linked to this article.

## Data availability

The data supporting the findings of this study are available from the corresponding authors upon request. RBNS sequencing data have been deposited at National Center for Biotechnology Information Sequence Read Archive and are publicly available using accession number PRJNA1185003. The source data underlying Figs. 1c, 2b–c, and 5b, Supplementary Figs. 2b and 5c–d are provided as a Source Data file. Source data are provided with this paper.

## Code availability

This study did not generate new code. To estimate $K_D$ values, we used the code previously published[25] (https://figshare.com/articles/software/MicroRNA-binding_thermodynamics_and_kinetics_by_RNA_Bind-n-Seq/19180952).

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

## Acknowledgements

We thank members of the Zamore and Jouravleva laboratories for critical comments on the manuscript. We thank the KYOTO *Drosophila* Stock Center (*Drosophila* Genomics and Genetic Resources, Kyoto Institute of Technology) for providing fly stocks used in this study. This work was supported in part by National Institutes of Health grant R35 GM136275 to P.D.Z. and by ATIP-Avenir funding (CNRS Biology) to K.J.

## Author contributions

Conceptualization, J.V.B, K.J., and P.D.Z.; Methodology, J.V.B. and K.J.; Investigation, J.V.B. and K.J.; Formal analysis, K.J. and J.V.B.; Writing—original draft, K.J. and J.V.B.; Writing—review and editing, K.J., J.V.B., and P.D.Z.; Supervision, K.J. and P.D.Z.; Funding acquisition, K.J. and P.D.Z.

## Competing interests

The authors declare no competing interests.
