## [Transparent Peer Review file · Nature Communications]

Biochemical principles of miRNA targeting in flies

Corresponding Author: Dr Karina Jouravleva

Version 0:

Reviewer comments:

Reviewer #1

(Remarks to the Author)

The authors performed RNA Bind-n-Seq (RBNS) on four miRNAs associated with fly Ago1. Using these data they identified sequences corresponding to ≤ 10 nt binding sites of these four miRNAs and reported their binding affinities. They concluded that, compared to mammals, fly miRNAs have a narrower range of binding-site diversity, and that some noncanonical site types originally defined in mammals (nucleation-bulged and 3'-only sites) also function in flies.

Major concerns:

1) The ability to estimate absolute Kd values from RBNS data is intriguing, but difficult to understand as described. Based on ref. 23, the authors appear to have sampled 100 different combinations of initial values for the AGO-miRNA stock concentration (in the range 0.5–25 nM) and the Kd value for the no-site sequences (in the range 0.5–10 nM). For each of these combinations, the authors let the parameters converge by minimizing the cost function to obtain a best fit for the estimated Kd of each enriched site type, the background, and the stock concentration of AGO-miRNA. Then, these values were combined and estimates from independent assays were merged. To help readers understand this last step, the authors should provide additional information. First, they should provide a supplemental table listing for each of the 100 combinations the fitted values (Kd values for enriched sites, the background, the stock concentration of AGO-miRNA, the no-site Kd value, and the cost-function value) at convergence. These should be provided for at least two miRNAs (let-7a and miR-184) and PUM2. Second, the authors should more clearly describe how these fitted values were combined and analyzed to arrive at the final Kd estimate. Currently the manuscript says, "To estimate KD values, run the code²³ by following instructions in the README file." However, the README file only lists a command without explaining how the Kd's were combined to arrive at a final estimate.

2) Fig. 1C. The analysis in Fig. 1C does not support the conclusion that canonical binding can be predicted by free energy of pairing. A more informative analysis would compare the relationship between measured Kd and predicted Kd for different representatives of the same site type. By combining data for 7nt sites with those for 6nt sites, the apparent significance of the relationship between measured Kd and predicted Kd evaluated in Fig. 1C is partially driven by the length of seed pairing rather than the predicted strength of seed pairing, raising the possibility that this length component is needed to achieve statistical significance. Because the importance of the length of seed pairing has been observed many times, a novel conclusion would require an analysis that avoids this confounding component. Indeed, when looking at the correlation across different miRNAs for the same site type, the predicted Kd values do not align with empirical Kd values. For example, the empirical Kd for the 7mer-m8 site of miR-184 is two-fold weaker than that of let-7, while the predicted Kd for the 7mer-m8 site of miR-184 is 25-fold stronger than that of let-7. Another problem with including different sites of the same miRNA in the same analysis is that the Pearson correlation calculation requires that the data points be independent, which is invalid for the combined data, as one would expect sites from the same miRNA to behave similarly to each other across site types. Fig. 5d and the associated analysis on 3'-only sites (page 10) have the same issues.

The comparison to previous results from analogous analyses for mammalian AGO2-miRNAs is also unclear. In the first sentence of the section the authors cite mammalian studies showing that "[miRNAs] with strong predicted free energy of site pairing bind their target sites with higher affinity than miRNAs with weak seed-pairing" but that "the difference in binding affinities of mammalian miRNAs is less than might have been expected." Doesn't the slope of the line in Fig. 1C, which is much less than 1:1, suggest that the same is true in flies? To make a clearer comparison with mammals, the values for mammals and flies should both be plotted and each slope compared quantitatively to the 1:1 line.

3) Fig. S1a. The plotted values of Fig. S1a do not agree with the reported Kd values: the Kd value of the seed-matching

target appears to be around 60 pM instead of 5 pM, and that of the fully complementary target (based on the abnormally shaped sigmoidal curve) appears to be around 30 pM instead of 2 pM.

How do the authors reconcile the K_d value determined for the let-7 8mer Fig. S1a (~60 pM) with the value determined by RBNS in Fig. 1B (2 pM).

Also, the analysis of Fig. S1a may not support the conclusion that there is “no substantive difference in affinity between a seed-matching and a fully complementary target.” The conclusion of no substantive difference requires that the binding reaction(s) have reached equilibrium, which may not be the case for the fully paired target. Do the authors know the dissociation rate of the fully complementary target? If so, was the incubation time of experiment described in Fig. S1a (2 hours) sufficient to achieve binding equilibrium?

4) Fig. S2-3. The authors use AlphaFold3-predicted structures to investigate seed-mismatch tolerance of miR-184. This is not a proper use of AlphaFold. AlphaFold3 fails to recognize RNA mismatches in AGO due to heavy template matching to existing AGO structures, which all feature perfectly matched seed sequences, and thus shoehorns mismatched nucleotides across each other as if they were complementary. As the authors noted, these predicted structures add no value to this work or its conclusions; they should be removed from the paper.

5) A central and provocative claim of this manuscript is that the repertoire of miRNA binding sites is less diverse for fly Ago1-miRNAs than for mammalian AGO2-miRNAs. However, the evidence for this claim is not clearly presented. Only one miRNA (let-7) is shared between the fly and mammalian RBNS datasets. Are the differences observed for other miRNAs due to inter-species differences or differences between different miRNA sequences tested? It would also help to know what metric the authors are comparing. Are the authors saying that fly Ago1 has fewer unique sites in the de novo site identification? This metric would be sensitive to differential noisiness of the data. Or, are there fewer sites with affinities within X-fold of the 8mer site, or fewer sites more than X-fold better than no-site sequences? The metric being used should be stated and the associated value declared for each mammalian miRNA and each fly miRNA. Then, the distribution of values for the fly miRNAs should be compared to the distribution for the mammalian miRNAs, and a statistical test should be performed to evaluate if the difference between species is more than that expected based on the distribution within each species.

The authors claim that fly Ago1-miRNAs have a lower tolerance for seed-region imperfections than mammalian AGO2-miRNAs, which they invoke as a possible explanation for their more general claim (discussed above) that fly Ago1-miRNAs have less diverse repertoires of binding sites. A more quantitative comparison is needed to support this claim. The authors should compare the ΔG penalties associated with individual seed region imperfections, aggregating across the four fly miRNAs from this study. They should then compare these to the ΔG penalties for the mammalian miRNAs with RBNS datasets. If the penalties associated with a particular class of pairing imperfection (e.g. mismatches) are globally/generally larger for the fly miRNAs compared to the mammalian miRNAs, and if these differences are significantly more prominent between species than within species, then it would be valid to argue that fly Ago1 has lower tolerance for that class of pairing imperfection.

6) The analysis of nucleation-bulge sites should be improved. Nucleation-bulge sites are also 6mer-A1 sites. Therefore, the null hypothesis should be that the nucleation-bulge sites bind at the same affinity as the typical 6mer-A1 sites, and evidence for the function of the nucleation-bulge would come from observation of greater affinity than that observed for other 6mer-A1 sites with similar flanking nucleotide composition. By these criteria, only some mammalian miRNAs recognize nucleation-bulge sites. Is the same true in flies?

Minor concerns:

1) page 3. Refs 1–7 are cited in a way that suggests that some references support mRNA degradation and others support translational inhibition. However, most of them report a combination of mRNA degradation and translational inhibition, and so they might best be cited together. When looking at the relative contributions of mRNA degradation and translational inhibition, ref. 3 does not distinguish between mRNAs that have a site to the miRNA and those that do not. Therefore, ref. 3 seems less relevant than the others for this citation.

2) page 3. Because central sites do have some seed pairing that might still function to nucleate pairing, it might be best to not characterize them as “seedless.”

3) page 3. “The most successful computational approaches typically predict miRNA binding rather than biologically important regulation ref 9,22,25.” Actually, these approaches primarily predict site efficacy (not site binding). Some also provide the option of predicting site conservation, which is one approach for predicting biologically important regulation. For example, the approach of reference 22 is trained to predict site efficacy as well as site binding. The approach of reference 25 is trained to predict site efficacy, with no training to predict site binding affinity. It also provides the option of considering only site conservation. (Note that the approaches that predict site efficacy also consider site conservation to the extent that more conserved sites have greater efficacy.)

4) pages 3-4. The purpose of mentioning the lin-4:lin-14, bantam:hid, bantam:clock and miR-9:TLX interactions is unclear. Is it to support the previous sentence, which states that prediction of animal regulatory targets is difficult? If so, these examples do not all serve this purpose. The top predicted target of lin-4 is lin-14, with lin-28 (another genetically identified lin-4 target) tied for second (TargetScanWorm). Likewise, hid is either the top predicted target of bantam or the third-highest predicted target, depending upon whether the target sites are ranked based on evolutionary conservation or predicted site efficacy

(TargetScanFly).

- 5) To put the weaker Kd values into perspective, the no-site Kd values should also be plotted in Fig. 1b.
- 6) The 10,000 mark of the x axis in Fig. 1c needs to be corrected.
- 7) To better explain RBNS analysis and illustrate the quality of the new data, Fig. 1 should also have a panel showing how binding enrichment varies over fly Ago1-miRNA concentrations (as in Fig. 3B of ref. 23) and also showing the fits to the enrichments (as in Fig. 1E of ref. 22).
- 8) When comparing results from fly and mammalian RBNS studies, the authors should note that the fly study used 3.5 mM Mg²⁺, whereas the original mammalian study used 1 mM Mg²⁺.
- 9) page 11. "These data suggest that Drosophila Ago1 competes with local RNA secondary structures for site binding." The authors could mention that similar results have been reported when analyzing RBNS data for mammalian AGO2.
- 10) page 12. "A reporter assay using Drosophila S2 cells monitored changes in steady-state levels of all cellular mRNAs ...". It is unclear what the "reporter assay" is referring to in this sentence.

Reviewer #2

(Remarks to the Author)

The codes of target genes recognition by mature miRNA sequences are very important for miRNA-mediated gene regulation and the binding affinity between miRNA and target sequences greatly determines how strong the target genes can be repressed by miRNAs. The biochemical basis for mammalian miRNA interaction with targets has been well studied while the biochemical rule for fly Ago targeting is still not fully known. In this manuscript "Biochemical principles of miRNA targeting in flies", Vega-Badillo et al systematically investigate the binding rules of fly Ago1-miRNA complex with RNA targets using RNA Bind-n-Seq (RBNS) assay. Although most of the targeting rules are similar with mammalian miRNAs which are already known, they still find some different targeting features, for example, fly miRNAs have a narrow binding diversity which favors perfect matched seed sites with limited tolerance for seed imperfections. They also identified 3'-only sites and nucleation-bulged sites, but no central paired sites are found due to the cleavage activity of fly Ago1 on these sites.

This study makes up the gap for biochemical basis of fly miRNA-target interactions and provides prediction guidance for fly miRNA mediated gene regulation in the future. I would recommend this manuscript for publication in Nature Communications if the following concerns can be addressed.

1. As similar studies have been done by David Bartel lab (Ref #22 and #33) in mammalian system using same methods, the authors should provide a summary figure or table to compare the similarities and differences for the targeting rules between fly and mammal miRNAs, and discuss the potential reasons for the differences in this summary. This will help the readers to easily understand the targeting rules in different species.
2. In this study, the authors tested four different fly miRNAs on their targeting rules and showed some differences on their target binding behaviors. Can the authors summarize their major binding differences somewhere in the manuscript and explain the potential reasons why different miRNA sequences can cause different binding activities?
3. In Fig 1B, the authors showed binding activities of 6mer targets (better than 6mer-m8 and 6mer-A1), which is interesting to know if these sites are really functional in vivo. Can the authors do some sensor tests for these 6mer sites to see if they can mediate detectable regulation?
4. The central paired sites are not identified in this study due to Ago1 mediated cleavage of these sites as shown in Fig 5b. Can these central paired sites (without seed match) really mediate detectable target cleavage in cells because it's presumably not involving GW182? Or these central paired sites can mediate target repression via normal miRNA way which is GW182 dependent? If these are real regulatory sites, it may encourage others to annotate such sites.

Minor concerns:

5. In Fig 1A, please indicate that the RNA oligo pool (input) is also sequenced.
6. In Fig 1A, the blocking oligonucleotide is DNA while the sequences are showed as RNA with "U".
7. In Fig 1C, there is a typo in X-axis-the last "100" should be "10,000".
8. In this sentence "In plants and animals, ~22-nt-long microRNAs (miRNAs) guide AGO-clade Argonaute proteins to repress partially complementary mRNA targets by accelerating their degradation 1–3 or inhibiting their translation 4–7." As known, in plants, miRNAs are most perfectly matched to target RNAs and guide their cleavage in an RNAi manner.
9. Please unify the citation format. "miRNAs display sequence-specific differences in binding their canonical sites: those with strong predicted free energy of site pairing bind their target sites with higher affinity than miRNAs with weak seed-pairing 21 ; McGeary et al., 2019, #16; Jouravleva et al., 2022, #143956}."
10. Typos in these sentences: "To study the energetics of canonical binding of fly miRNAs, we loaded recombinant Ago1

loaded with one of four different miRNAs”; “Our data suggest that Ago1 may bind sites containing one single central mismatch but identifying universal rules for predicting which miRNAs can bind 8mer-x4R sites with unexpectedly high affinity high-throughput target binding measurements for a larger set of miRNAs”

Reviewer #3

(Remarks to the Author)

The manuscript by Vega-Badillo et al. describes the biochemical exploration of in vitro miRNA binding in the context of *Drosophila* Ago1. Building upon previous use of this technology for mammalian Ago2, the authors describe distinctions in *Drosophila* Ago1-mediated miRNA interactions that may help future predictive efforts in *Drosophila*. In general, the experiments are well-done and well-described, and generates useful insights for the fly miRNA community to further explore the mechanisms of how miRNAs regulate various aspects of biology in that system.

I only really have minor comments:

“At these positions mismatched nucleotides A and G are often the least tolerated, likely because their purine rings are larger than pyrimidines and thereby cause more steric hindrance.” – can the authors provide some quantification of “often”? It seems true for let-7, but somewhat less for the others – I think just having some number here would be clearer than a vague ‘often’

“Our data suggest that Ago1 may bind sites containing one single central mismatch but identifying universal rules for predicting which miRNAs can bind 8mer-x4R sites with unexpectedly high affinity high-throughput target binding measurements for a larger set of miRNAs.” – this sentence seems like it’s missing an ending (‘our data suggest ... but identifying universal rules for ... [will be impossible? Will require more extensive profiling?]’)

The analysis in Fig 6 is interesting, but it might be good to show separately in supplementary figures the 5’ and 3’ end versions, since the 5’ side is also impacted by whether the sequence base-pairs with the miRNA. I’m also a bit confused by the conclusion of the Fig 6 section – is the overall argument in this section that secondary structure has been shown to influence repression in [[emphasis mine]]mammalian]] cells, and this section validates the same principle for *Drosophila* Ago1? Or that RBNS scores can essentially serve as a functional proxy that correlates with (and thus likely incorporates the effect of) local structure? I don’t think it’s clear from the manuscript whether the same finding is true (& has already been shown) for RBNS AGO2 data in human, or just the general principle

I appreciate that the authors haven’t tried to over-sell the impact of this work, but I think having a little more discussion of the impact of this work would be useful – I can see the obvious impact in aiding *Drosophila*-specific prediction tools, but I think having a touch more insight into what else the authors envision of the impact would be great

Version 1:

Reviewer comments:

Reviewer #1

(Remarks to the Author)

The authors have satisfactorily addressed nearly all of my concerns. I just have a few remaining suggestions/clarifications.

Regarding the last part of major concern 5, I apologize that my initial comments were unclear, but I was requesting that the authors show the deltaG penalties calculated from the experimental values (as calculated from RBNS-derived Kd values by the equation $\Delta G = RT \ln(Kd)$), rather than the deltaG values predicted from the nearest-neighbor parameters. Showing the experimental values in Figure S4 should speak to the claim that fly Ago1 is less tolerant of seed-region imperfections than human AGO2.

With respect to minor concern 1, the current model for miRNA-mediated repression centers on miRNA-mediated recruitment (through TNRC6) of a deadenylase complex (PAN2/PAN3 or CCR4-NOT) to accelerate mRNA poly(A) tail shortening, which in most settings enhances mRNA decay by hastening the time at which the tail becomes too short to resist mRNA decapping. In addition, the CCR4-NOT deadenylation complex brings to the mRNA the proteins responsible for translational inhibition of the mRNA (i.e., 4EHP and 4E-T; PMID28487484 from Sonenberg lab). Because the proteins that shorten the tail and the proteins that inhibit translation presumably arrive concurrently, it seems likely that the early step in mRNA decay (i.e., accelerated deadenylation) is happening in parallel with (and at the same time as) the effects on translation. These more recent mechanistic insights are not consistent with the earlier proposals that mRNA decay is a secondary effect of translational repression; for these earlier proposals to be plausible, translational repression would need to occur before accelerated poly(A) tail shortening (which does not appear to be the case in the settings examined). I hope this clarifies the concept that miRNAs mediate a combination of mRNA degradation and translational repression.

One more clarification: Regarding the last part of major concern 2 (which the authors found puzzling “because the original version of the manuscript stated that flies and mammals behaved similarly”), my concern was not with the quoted text but with a statement in the original version of the text, saying, “Unlike in mammals, these differences can be predicted by existing models of RNA duplex stability in solution.”

Reviewer #2

(Remarks to the Author)

I have no further comments and support the publication of this work in Nature Communications.

Reviewer #3

(Remarks to the Author)

The authors have addressed my comments and I support publication

We thank the Reviewers for their interest in our work and for their thoughtful and constructive feedback. We address each point below.

Reviewer #1

Reviewer 1 raises many useful points, many of which underscore that we need to do a better job educating our readers in the fundamentals of nucleic acid binding kinetics and equilibria. Their comments also helped us improve some of our computational analyses. However, we were surprised by the reviewer's aggressive tone and would like to remind the reviewer that it would have been possible to express all of these points without being condescending. A trainee is on the receiving end of every manuscript review, and the adversarial review process is a significant factor in convincing young scientists to abandon academia.

The authors performed RNA Bind-n-Seq (RBNS) on four miRNAs associated with fly Ago1. Using these data they identified sequences corresponding to ≤ 10 nt binding sites of these four miRNAs and reported their binding affinities. They concluded that, compared to mammals, fly miRNAs have a narrower range of binding-site diversity, and that some noncanonical site types originally defined in mammals (nucleation-bulged and 3'-only sites) also function in flies.

Major concerns:

*1) The ability to estimate absolute K_d values from RBNS data is intriguing, but difficult to understand as described. Based on ref. 23, the authors appear to have sampled 100 different combinations of initial values for the AGO-miRNA stock concentration (in the range 0.5–25 nM) and the K_d value for the no-site sequences (in the range 0.5–10 nM). For each of these combinations, the authors let the parameters converge by minimizing the cost function to obtain a best fit for the estimated K_d of each enriched site type, the background, and the stock concentration of AGO-miRNA. Then, these values were combined and estimates from independent assays were merged. To help readers understand this last step, the authors should provide additional information. First, they should provide a supplemental table listing for each of the 100 combinations the fitted values (K_d values for enriched sites, the background, the stock concentration of AGO-miRNA, the no-site K_d value, and the cost-function value) at convergence. These should be provided for at least two miRNAs (*let-7a* and *miR-184*) and *PUM2*. Second, the authors should more clearly describe how these fitted values were combined and analyzed to arrive at the final K_d estimate. Currently the manuscript says, "To estimate K_D values, run the code²³ by following instructions in the README file." However, the README file only lists a command without explaining how the K_d 's were combined to arrive at a final estimate.*

We thank the Reviewer for pointing out that the Material and Method section insufficiently described our procedure to estimate K_D values from RBNS datasets. We edited the relevant paragraph to read,

“To estimate K_D values, run the code (Jouravleva et al., 2022) by following instructions in the README file. Bootstrapping of 95% of the data was performed ten times on sequencing reads from each binding reaction and the RNA pool. MLE of K_D values was performed on each bootstrapped sample by using 100 different combinations of 10 initial guesses of miRISC concentration (in the range 0.5–25 nM) and 10 initial guesses of K_D for RNA with no enriched site (in the range 0.5–10 nM). K_D values were initialized as the inverse of the average enrichment values. The background was initialized at 0.1 nM. All the initial guesses were partially randomized by adding a value drawn from a normal distribution with mean 0 and standard deviation 0.1. The cost function was evaluated in the presence of physically meaningful constraints on the parameters: $0.1 \text{ pM} \leq K_D^{\text{site}} \leq 100 \text{ nM}$, $100 \text{ pM} \leq K_D^{\text{no-site}} \leq 10,000 \text{ nM}$, $100 \text{ pM} \leq \text{miRISC} \leq 100 \text{ nM}$, and $5 \text{ pM} \leq \text{background} \leq 5 \text{ nM}$. Any of the fitted parameters were at the boundaries at the end of the optimization routine. K_D estimates, the background, and the stock concentration of miRISC provided by MLE were used to predict counts of each binding site type in sequencing data. These counts were compared with observed sequencing data, and MLE results were retained if Pearson correlation coefficient was >0.90 . Results from independent starting points satisfying this criterion were combined. All bootstrapped samples were combined. Finally, estimates from two independent RBNS assays were merged. Median and 95% confidence intervals on medians were reported.

“To estimate K_D values of PUM2, fastq files of ENCFF761JAF, ENCFF894MLG, ENCFF090QVU, ENCFF574TBN, ENCFF266PHU, and ENCFF345UJY datasets were downloaded from the ENCODE project. Reads containing one single PUM binding site—UGUAUAUA (consensus binding site), AGUAUAUA, CGUAUAUA, and GGUAUAUA—were identified. Reads containing multiple instances of the PUM binding sites were not included in the analysis and represented $\leq 0.001\%$ of libraries. Reads that did not have any of the above binding motifs were classified as reads with a no-site. MLE of PUM2 K_D values was performed as described for miRISC, except: (1) 100 different combinations of 10 initial guesses of PUM2 concentration (in the range 0.5–2.3 μM) and 10 initial guesses of K_D for RNA with no enriched site (in the range 1–500 nM) were used, and (2) the cost function was evaluated in the presence of physically meaningful constraints on the parameters: $0.1 \text{ pM} \leq K_D^{\text{site}} \leq 1 \mu\text{M}$, $100 \text{ pM} \leq K_D^{\text{no-site}} \leq 1 \mu\text{M}$, $100 \text{ pM} \leq \text{PUM2} \leq 5 \mu\text{M}$, and $5 \text{ pM} \leq \text{background} \leq 500 \text{ nM}$.”

We also now provide fitted estimates at convergence for PUM2 and canonical sites of the five miRNAs used in this study (Table S1).

2) Fig. 1C. The analysis in Fig. 1C does not support the conclusion that canonical binding can be predicted by free energy of pairing. A more informative analysis would compare the relationship between measured K_d and predicted K_d for different representatives of the same site type. By combining data for 7nt sites with those for 6nt

sites, the apparent significance of the relationship between measured K_D and predicted K_D evaluated in Fig. 1C is partially driven by the length of seed pairing rather than the predicted strength of seed pairing, raising the possibility that this length component is needed to achieve statistical significance. Because the importance of the length of seed pairing has been observed many times, a novel conclusion would require an analysis that avoids this confounding component. Indeed, when looking at the correlation across different miRNAs for the same site type, the predicted K_D values do not align with empirical K_D values. For example, the empirical K_D for the 7mer-m8 site of miR-184 is two-fold weaker than that of let-7, while the predicted K_D for the 7mer-m8 site of miR-184 is 25-fold stronger than that of let-7. Another problem with including different sites of the same miRNA in the same analysis is that the Pearson correlation calculation requires that the data points be independent, which is invalid for the combined data, as one would expect sites from the same miRNA to behave similarly to each other across site types. Fig. 5d and the associated analysis on 3'-only sites (page 10) have the same issues.

We respectfully disagree with the assertion that the observed relationship is primarily driven by seed length. The nearest-neighbor rules predict that the additional base in longer sites is indeed paired, and therefore the free energy of pairing should remain informative for canonical binding. Nonetheless, we recognize that separating the site types, while addressing the Reviewer's concern, markedly reduced statistical power and limited the strength of the analysis. In the interest of clarity and to avoid an extended debate on this point, we removed this analysis from the revised manuscript.

The comparison to previous results from analogous analyses for mammalian AGO2-miRNAs is also unclear. In the first sentence of the section the authors cite mammalian studies showing that “[miRNAs] with strong predicted free energy of site pairing bind their target sites with higher affinity than miRNAs with weak seed-pairing” but that “the difference in binding affinities of mammalian miRNAs is less than might have been expected.” Doesn't the slope of the line in Fig. 1C, which is much less than 1:1, suggest that the same is true in flies? To make a clearer comparison with mammals, the values for mammals and flies should both be plotted and each slope compared quantitatively to the 1:1 line.

We were puzzled by this comment, because the original version of the manuscript stated that flies and mammals behaved similarly:

“These results mirror those for mammalian miRNAs. Moreover, direct binding measurements found no substantive difference in affinity between a seed-matching ($K_D = 5 \pm 3$ pM) and a fully complementary target ($K_D = 2 \pm 1$ pM; Fig. S1a), suggesting that seed complementarity dominates *Drosophila* Ago1 target binding, just like mammalian AGO2.” (line 119ff in the revised text)

The text states that both mammals and flies showed smaller differences in binding affinities than might have been expected, and we drew the same conclusion as the reviewer.

3) Fig. S1a. The plotted values of Fig. S1a do not agree with the reported K_D values: the K_D value of the seed-matching target appears to be around 60 pM instead of 5 pM, and that of the fully complementary target (based on the abnormally shaped sigmoidal curve) appears to be around 30 pM instead of 2 pM.

The reviewer is incorrect in their analysis of the K_D values. The reported values of 5 pM and 2 pM are the correct values. Nor is the graph “abnormal”; in fact, it is the classical shape expected, which is not sigmoidal (Reviewers’ Figure 1, data plotted on a linear x-axis). However, the reviewer is not alone in making this error: Herschlag and colleagues (Jarmoskaite et al., *eLife* 2020; PMID 32758356) estimate that more than one-third of literature K_D values for RNA-protein interactions make the same mistake.

The assumption that $K_{1/2} = K_D$ is only true when $[R]_{\text{total}} \ll K_D$. When the K_D is in the pM range, this condition is impossible to satisfy using current methods of RNA labeling. Thus, one must instead use the quadratic binding equation, i.e., the explicit solution to the binding equation.

We used 0.1 nM RNA target, so $[R]_{\text{total}} \gg K_D$. In this case, the concentration of P that gives half binding does not equal to the K_D and is instead described by Equation 1, below:

$$f = \frac{([E_T] + [S_T] + K_D) - \sqrt{([E_T] + [S_T] + K_D)^2 - 4[E_T][S_T]}}{2[S_T]}$$

where f is the fraction target bound, $[E_T]$ is the total RISC concentration, $[S_T]$ is the total RNA target concentration and K_D is the equilibrium dissociation constant. Under such a “titration” regime, the data *should not be sigmoidal when plotted on a log x-axis*. Herschlag and coworkers provide an illustration of

[RNA target] = 0.1 nM

Reviewers’ Figure 1 | Linear plots of the same binding data in Fig. S1a. The curves show the expected shape for very tight binding measured under titration conditions.

this in their Figure 5, figure supplement 1, and have an extensive discussion on distinguishing between concentration regimes. Moreover, we described our fitting method and noted that our experiments were performed under a “titration” regime (intermediate regime) in the Methods of our manuscript. We have now highlighted this further by adding to Figure 1c in the revised manuscript the concentrations of RISC and RNA, as well as the equation used to fit the data.

How do the authors reconcile the K_D value determined for the let-7 8mer Fig. S1a (~60 pM) with the value determined by RBNS in Fig. 1B (2 pM).

Please see above; the K_D for let-7 7mer-m8 in Figure S1a (Fig. 1c in the revised manuscript) is correct; the reviewer’s assumption that it is 60 pM is erroneous.

Also, the analysis of Fig. S1a may not support the conclusion that there is “no substantive difference in affinity between a seed-matching and a fully complementary target.” The conclusion of no substantive difference requires that the binding reaction(s) have reached equilibrium, which may not be the case for the fully paired target. Do the authors know the dissociation rate of the fully complementary target? If so, was the incubation time of experiment described in Fig. S1a (2 hours) sufficient to achieve binding equilibrium?

Yes. This topic is discussed in our recent paper (Jouravleva et al., *Cell Reports Methods* 2022) and in earlier work from Herschlag and colleagues (Jarmoskaite et al., *eLife* 2020). The time for a binding reaction to achieve equilibrium, T , is typically considered the time to reach 96.6% completion, which is described by Equation 2:

$$T = 5 \times t_{1/2} = 5 \times \ln 2 / k_{eq}$$

For the binding equilibrium where RISC interacts with an RNA site_{*i*}, the equilibration rate constant is described by Equation 3,

$$k_{eq} = k_{on} \times [RISC] + [site]_i + k_{off}$$

For let-7 bound to AGO2 and seed-matched target RNA, the seed determines k_{on} , so

$$k_{on} = 3 \times 10^7 \text{ M}^{-1}\text{s}^{-1} \text{ at } 25^\circ\text{C} \text{ (our unpublished data)}$$

In Fig. S1a, [RISC] varied between 0.0008 nM and 0.8 nM and the [RNA] was 0.1 nM. For a fully complementary target and let-7 bound to mouse AGO2, the homolog of fly Ago1,

$$k_{off} = 7.7 \times 10^{-4} \text{ s}^{-1} \text{ at } 25^\circ\text{C} \text{ (Wee et al., Cell 2013)}$$

Using these parameters, we can calculate the time required to achieve 96.6% completion in the approach to equilibrium:

$$k_{eq} = (3 \times 10^7 \text{ M}^{-1}\text{s}^{-1})(8 \times 10^{-12} \text{ M} + 1 \times 10^{-10} \text{ M}) + 7.7 \times 10^{-4} \text{ s}^{-1}$$

Returning to Equation 2, one can see that our experiments were explicitly designed to allow the reactions to exceed 96.6% completion in the approach to equilibrium:

$$T = 5 \times \ln 2 / (4.0 \times 10^{-3} \text{ s}^{-1}) = 14.4 \text{ min}$$

To demonstrate this empirically, we repeated the equilibrium binding assays in Fig. S1a (Fig. 1c in the revised manuscript) using a 16 h incubation time. The results for 2 h and 16 h are indistinguishable:

Site type	K_D , 16 h incubation	K_D , 2 h incubation
Seed-match	$7 \pm 3 \text{ pM}$	$5 \pm 3 \text{ pM}$
Fully complementary	$1 \pm 1 \text{ pM}$	$2 \pm 1 \text{ pM}$

We note that even in the limit case when both the RISC and site concentrations are infinitely small (i.e., approach zero), $k_{eq} = k_{off}$ so $T = 5 \times \ln 2 / k_{off} = 1.25 \text{ hours}$

4) Fig. S2-3. The authors use AlphaFold3-predicted structures to investigate seed-mismatch tolerance of miR-184. This is not a proper use of AlphaFold. AlphaFold3 fails to recognize RNA mismatches in AGO due to heavy template matching to existing AGO structures, which all feature perfectly matched seed sequences, and thus shoehorns mismatched nucleotides across each other as if they were complementary. As the authors noted, these predicted structures add no value to this work or its conclusions; they should be removed from the paper.

We overlooked the significant change between AlphaFold 2 and AlphaFold 3. We have removed the data from the paper. Thank you for catching our error.

5) A central and provocative claim of this manuscript is that the repertoire of miRNA binding sites is less diverse for fly Ago1-miRNAs than for mammalian AGO2-miRNAs. However, the evidence for this claim is not clearly presented. Only one miRNA (*let-7*) is shared between the fly and mammalian RBNS datasets. Are the differences observed for other miRNAs due to inter-species differences or differences between different miRNA sequences tested? It would also help to know what metric the authors are comparing. Are the authors saying that fly Ago1 has fewer unique sites in the de novo site identification? This metric would be sensitive to differential noisiness of the data. Or, are there fewer sites with affinities within X-fold of the 8mer site, or fewer sites more than X-fold better than no-site sequences? The metric being used should be stated and the associated value declared for each mammalian miRNA and each fly miRNA. Then, the distribution of values for the fly miRNAs should be compared to the distribution for the mammalian miRNAs, and a statistical test should be performed to evaluate if the difference between species is more than that expected based on the distribution within each species.

We agree with the Reviewer that de novo identification of binding sites based solely on enrichment values above an arbitrarily defined threshold (McGeary et al., 2019) could be affected by variable levels of noise in the data. However, we did not employ this approach: we instead define a motif as a binding site if its enrichment is greater than a chosen Z-score threshold (Lambert et al., 2014; Dominguez et al., 2018; Van Nostrand et al., 2020; Jouravleva et al., 2022). Specifically, a motif was considered significant if its Z-score was ≥ 99.9 percentile and was not enriched in the no-RISC control reaction (line 555ff in the revised text). This procedure identified similar binding sites in RBNS datasets

with different levels of noise generated with a double-filter or pull-down step (Figure S2C in Jouravleva et al., 2022). By contrast, a de novo site discovery algorithm that relied on an enrichment threshold rather than a Z-score threshold showed reduced sensitivity for detecting low-affinity sites (Figure S3D in Jouravleva et al., 2022).

To further test whether the repertoire of miRNA binding sites is less diverse for fly Ago1 than for mammalian AGO2, we analyzed an additional RBNS dataset for miR-124-loaded Ago1. We chose miR-124 because (1) its sequence is identical in flies and humans, and (2) the binding preferences of human miR-124-loaded AGO2 have been extensively characterized by RBNS (McGeary et al., 2019). In that study, human AGO2 bound 42 distinct site types with affinities ranging from 0.002 to 0.2 relative to the no-site K_D (Figure 2C in McGeary et al., 2019). We do not report affinities of fly Ago1 relative to no-site K_D , because the definition of “no-site” motifs introduces a bias: both our study and McGeary et al. define “no-site” motifs as all sequences except those for which K_D values are explicitly estimated. For example, the let-7 8mer-x4G is categorized as a no-site motif in our Figure 1b but not in Figure 3a.

Comparing K_D relative to the canonical 8mer site avoids this ambiguity. Using this normalization, human miR-124-loaded AGO2 binds ~15 site types within a 10-fold range of the 8mer affinity, spanning canonical and noncanonical interactions (including seed-matched sites with wobble pairs, mismatches, bulges, and 3'-only sites). In contrast, our de novo site discovery algorithm, which applies a Z-score threshold as described above, identified only three binding sites for fly miR-124-loaded Ago1: the canonical 8mer, 7mer-m8, and 7mer-A1 (Figure 5a). All other site categories, including those reported by McGeary et al. for human AGO2, showed affinities more than 10-fold weaker than that of fly 8mer (Figures S2a and S3b), with the exception of the 8mer-w5 site mentioned in the section on G:U pairing.

We revised our main text to add the miR-124 data:

“Finally, our de novo site discovery algorithm identified only three binding sites for fly miR-124-loaded Ago1: the canonical 8mer, 7mer-m8, and 7mer-A1 (Fig. 5a). Remarkably, human miR-124, whose sequence is identical to that in flies, loaded into AGO2 binds ~15 site types within a 10-fold range of the 8mer affinity, spanning canonical and noncanonical interactions (including seed-matched sites with wobble pairs, mismatches, bulges, and 3'-only sites) (McGeary et al., 2019). The ΔG penalties associated with individual seed region imperfections across the five fly miRNAs from this study are comparable to the ΔG penalties for the mammalian miRNAs from previous AGO2 RBNS datasets (McGeary et al., 2019; Jouravleva et al., 2022) (Fig. S4). Together, our data suggest that fly Ago1 has lower tolerance for pairing imperfection.” (line 211ff in the revised text)

The authors claim that fly Ago1-miRNAs have a lower tolerance for seed-region imperfections than mammalian AGO2-miRNAs, which they invoke as a possible explanation for their more general claim (discussed above) that fly Ago1-miRNAs have

less diverse repertoires of binding sites. A more quantitative comparison is needed to support this claim. The authors should compare the ΔG penalties associated with individual seed region imperfections, aggregating across the four fly miRNAs from this study. They should then compare these to the ΔG penalties for the mammalian miRNAs with RBNS datasets. If the penalties associated with a particular class of pairing imperfection (e.g. mismatches) are globally/general larger for the fly miRNAs compared to the mammalian miRNAs, and if these differences are significantly more prominent between species than within species, then it would be valid to argue that fly Ago1 has lower tolerance for that class of pairing imperfection.

Thank you for this suggestion. We compare ΔG penalties in Fig. S4.

6) *The analysis of nucleation-bulge sites should be improved. Nucleation-bulge sites are also 6mer-A1 sites. Therefore, the null hypothesis should be that the nucleation-bulge sites bind at the same affinity as the typical 6mer-A1 sites, and evidence for the function of the nucleation-bulge would come from observation of greater affinity than that observed for other 6mer-A1 sites with similar flanking nucleotide composition. By these criteria, only some mammalian miRNAs recognize nucleation-bulge sites. Is the same true in flies?*

We thank the Reviewer for this suggestion. We computed K_D values for the nucleation bulge sites and for 6mer-A1 sites with similar flanking context. Then we performed bootstrap-based hypothesis testing: for each bootstrapped sample i , we computed the difference $d_i = K_D^{6mer-A1} - K_D^{nucleation\ bulge\ site}$. Medians of the difference and 95% confidence intervals of these differences (in pM) across the five miRNAs used in our study are indicated below.

miRNA	median	95% CI
let-7	5.25	[3.60; 7.11]
bantam	444	[429; 460]
miR-11	169	[163; 175]
miR-184	699	[690; 707]
miR-124	701	[692; 713]

Since the 95% confidence interval does not include 0, the difference is statistically significant at the 0.05 level.

We added these data to the manuscript:

Moreover, the nucleation-bulge sites across the five miRNAs tested bind at greater affinity than 6mer-A1 sites with similar flanking nucleotide composition (Fig. S3d). (line 188ff in the revised text)

Minor concerns:

1) *page 3. Refs 1–7 are cited in a way that suggests that some references support mRNA degradation and others support translational inhibition. However, most of them report a combination of mRNA degradation and translational inhibition, and so they*

might best be cited together. When looking at the relative contributions of mRNA degradation and translational inhibition, ref. 3 does not distinguish between mRNAs that have a site to the miRNA and those that do not. Therefore, ref. 3 seems less relevant than the others for this citation.

The concept that miRNAs mediate “a combination of mRNA degradation and translational repression” is puzzling. Perhaps we can start with some common ground? Can we agree that the molecules measured by ribosome footprinting have not been subjected to RNA degradation? If we can agree on that point, then perhaps we can persuade you that it is not possible for a single molecule of mRNA to be regulated by a combination of mRNA degradation and translational inhibition. Is the reviewer’s model that any given miRNA molecule might stochastically repress translation or trigger turnover on an individual mRNA molecule? Isn’t a simpler model that translational repression precedes mRNA turnover? Such a model predicts that a snapshot of miRNA-directed repression will be a mix of extant mRNAs undergoing translational repression and missing mRNAs that have already been destroyed.

Ref. 4 reports that translational repression precedes mRNA decay. The decay is a secondary effect caused by the translational repression, not the miRNA; at developmental stages when poly(A) tail shortening does not alter mRNA stability, mRNA decay plays a minor role at best in miRNA function. Thus, Ref. 4 supports translational repression. Hu and Collier (*Cell Research* 2012) expressed this succinctly:

Recent research results, however, suggest that in both zebrafish and fruit fly, translational inhibition is the initiating event of miRNA-mediated gene silencing.

We should have included the work of Cottrell et al. (*Science Adv* 2017), which showed that both 3’ UTR sequence and codon optimality can alter the effect size of miRNA-directed translational repression; we have now added that reference. Conversely, Ref. 2 reports that miRNAs act to trigger mRNA turnover *before* any effect on the rate of translation. Therefore, the lack of translation is a consequence of the mRNA no longer existing.

2) page 3. Because central sites do have some seed pairing that might still function to nucleate pairing, it might be best to not characterize them as “seedless.”

We have revised the text accordingly.

3) page 3. “The most successful computational approaches typically predict miRNA binding rather than biologically important regulation ref 9,22,25.” Actually, these approaches primarily predict site efficacy (not site binding). Some also provide the option of predicting site conservation, which is one approach for predicting biologically important regulation. For example, the approach of reference 22 is trained to predict site efficacy as well as site binding. The approach of reference 25 is trained to predict site efficacy, with no training to predict site binding affinity. It also provides the option of considering only site conservation. (Note that the approaches that predict site efficacy also consider site conservation to the extent that more conserved sites have greater efficacy.)

“Site efficacy” does not measure biologically important regulation. Because miRNAs work by binding their sites then recruiting other proteins that perform the actual regulation, measurement of changes in mRNA abundance in cultured cells mainly if not entirely reflects the efficiency of binding. Computational approaches do not measure biologically important regulation, which can only be tested by showing in vivo that mutation of a binding site or a deletion of a miRNA produces a *phenotype*. There is no evidence supporting the view that most miRNA-binding sites identified by TargetScan are important in vivo, and there is considerably evidence that just one or two targets account for the entirety of the observed phenotype when individual miRNAs are deleted in specific tissues or cell types in model organisms.

Of course, the reviewer is aware of the literature describing the high false-positive rate of current target prediction algorithms. We do not wish to engage in an extensive debate, but the two main lines of evidence are (1) the error of including effect sizes smaller than the inter-cell or inter-animal variation in mRNA abundance (Seitz, *RNA Biology* 2017; Pinzón et al., *Genome Res* 2017); and (2) including in conservation of sites in species lacking the miRNA (Pinzón et al., *Genome Res* 2017). The second objection has been rectified in more recent versions of TargetScan, but the first remains a matter of serious debate in the field. For a more comprehensive discussion of these and other issues with current methods of target prediction, please see Mockly and Seitz (*Methods Mol Biol* 2019).

Finally, doesn't Fig. 5C in McGeary et al. (*Science* 2019) suggest that K_D values alone outperform TargetScan 7 in predicting miRNA targets?

4) pages 3-4. The purpose of mentioning the *lin-4:lin-14*, *bantam:hid*, *bantam:clock* and *miR-9:TLX* interactions is unclear. Is it to support the previous sentence, which states that prediction of animal regulatory targets is difficult? If so, these examples do not all serve this purpose. The top predicted target of *lin-4* is *lin-14*, with *lin-28* (another genetically identified *lin-4* target) tied for second (TargetScanWorm). Likewise, *hid* is either the top predicted target of *bantam* or the third-highest predicted target, depending upon whether the target sites are ranked based on evolutionary conservation or predicted site efficacy (TargetScanFly).

Our purpose was quite different from the reviewer's guess. The examples we cited are among the few miRNAs whose biological function has been established by appropriate genetic tests in vivo. Of course, any algorithm that failed to predict the small number of biologically validated targets for specific miRNAs would not have been published, so the reviewer's argument reflects ascertainment bias, not any particular property of target prediction methods.

Unlike *Drosophila* siRNAs and piRNAs, miRNAs in general work by binding their sites, then recruiting other proteins that perform the actual regulation. Thus, it is critical to distinguish “predicted site efficacy,” which measures binding in cells, and biological function, which can only be measured in a living organism by performing two types of experiments: (1) deletion of the miRNA and characterizing the phenotype that results; and (2) mutation of the predicted

binding site in the proposed target. Needless to say, these are time consuming and laborious experiments.

In retrospect, we should have cast a wider net, since we neglected to cite the Großhans lab's elegant work in worms, which showed that *let-7* controls male and female sexual organ morphogenesis and skin progenitor cell fates by silencing a single target, *lin-41* (Aeschmann et al., *Life Sci Alliance* 2019; PMID 30910805). Similarly, the function of *let-7* in the vulval-uterine system was shown by mutation of *let-7* target sites to be explained solely by repression of *lin-41* (Ecsedi et al., *Dev Cell* 2015). These data argue strongly that current strategies for target prediction do not reflect biological function. We have now added these references to the manuscript.

5) *To put the weaker K_D values into perspective, the no-site K_D values should also be plotted in Fig. 1b.*

In our opinion, the definition of “no-site” motifs introduces a bias: both our study and McGeary et al. define “no-site” motifs as all sequences except those for which K_D values are explicitly estimated. For example, the *let-7* 8mer-x4G is categorized as a no-site motif in our Figure 1b but not in Figure 3a.

Operationally, no-site K_D values should not be compared among different miRNAs or different Argonaute proteins. We have explicitly measured the no-site K_D for fly Ago1 using a target containing purine ribonucleosides at every fourth position. Purine ribonucleoside contains no exocyclic groups and thus cannot pair with any natural base. Since three contiguous base pairs cannot form a stable helix, this target should not be able to bind any miRNA loaded into an AGO clade protein. To complement this approach, we also used a target consisting of 30 adenosines (A_{30}). The measured binding affinities for these sites are 3.7 ± 0.8 nM and 12 ± 3 nM, respectively. The new data are shown in Fig. S2b in the revised manuscript.

6) *The 10,000 mark of the x axis in Fig. 1c needs to be corrected.*

We fixed the typo. Thank you!

7) *To better explain RBNS analysis and illustrate the quality of the new data, Fig. 1 should also have a panel showing how binding enrichment varies over fly Ago1-miRNA concentrations (as in Fig. 3B of ref. 23) and also showing the fits to the enrichments (as in Fig. 1E of ref. 22).*

Thank you for this suggestion. We now show enrichment profile of canonical sites in RBNS datasets generated in this study in Figure S1.

8) *When comparing results from fly and mammalian RBNS studies, the authors should note that the fly study used 3.5 mM Mg^{2+} , whereas the original mammalian study used 1 mM Mg^{2+} .*

In response to the reviewer's concern, we directly tested the effect of high vs. low Mg^{2+} . We repeated the RBNS experiments at 0.89 mM Mg^{2+} , the concentration reported in the methods section of McGeary et al. As expected,

we recover fewer site-types at lower Mg^{2+} . We now include these data for *bantam* in the revised manuscript (Fig. S5a).

9) page 11. *“These data suggest that Drosophila Ago1 competes with local RNA secondary structures for site binding.” The authors could mention that similar results have been reported when analyzing RBNS data for mammalian AGO2.*

We edited the text accordingly.

10) page 12. *“A reporter assay using Drosophila S2 cells monitored changes in steady-state levels of all cellular mRNAs ...” It is unclear what the “reporter assay” is referring to in this sentence.*

Thank you for noting this imprecision in the text. We edited the relevant sentence to read, “RNA sequencing using *Drosophila* S2 cells monitored changes in steady-state levels of all cellular mRNAs after cell transfection with supraphysiological concentrations of exogenous miRNA duplexes.”

Reviewer #2 (Remarks to the Author):

The codes of target genes recognition by mature miRNA sequences are very important for miRNA-mediated gene regulation and the binding affinity between miRNA and target sequences greatly determines how strong the target genes can be repressed by miRNAs. The biochemical basis for mammalian miRNA interaction with targets has been well studied while the biochemical rule for fly Ago targeting is still not fully known. In this manuscript “Biochemical principles of miRNA targeting in flies”, Vega-Badillo et al systematically investigate the binding rules of fly Ago1-miRNA complex with RNA targets using RNA Bind-n-Seq (RBNS) assay. Although most of the targeting rules are similar with mammalian miRNAs which are already known, they still find some different targeting features, for example, fly miRNAs have a narrow binding diversity which favors perfect matched seed sites with limited tolerance for seed imperfections. They also identified 3'-only sites and nucleation-bulged sites, but no central paired sites are found due to the cleavage activity of fly Ago1 on these sites.

This study makes up the gap for biochemical basis of fly miRNA-target interactions and provides prediction guidance for fly miRNA mediated gene regulation in the future. I would recommend this manuscript for publication in Nature Communications if the following concerns can be addressed:

1. As similar studies have been done by David Bartel lab (Ref #22 and #33) in mammalian system using same methods, the authors should provide a summary figure or table to compare the similarities and differences for the targeting rules between fly and mammal miRNAs, and discuss the potential reasons for the differences in this summary. This will help the readers to easily understand the targeting rules in different species.

Thank you for this suggestion. We report this information in Table 1.

2. In this study, the authors tested four different fly miRNAs on their targeting rules and showed some differences on their target binding behaviors. Can the authors summarize

their major binding differences somewhere in the manuscript and explain the potential reasons why different miRNA sequences can cause different binding activities?

In our opinion, differences in canonical binding can be predicted by existing models of RNA duplex stability in solution. However, in the interest of clarity and to avoid an extended debate on this point with the Reviewer #1, we removed this analysis from the revised manuscript.

Regarding non-canonical binding by miRNA-loaded Ago1, the repertoire of these site types is limited in flies. Some site types can be productively bound by Ago1 in vitro, such as miR-184 8mer x4C; but identifying universal rules for predicting which miRNAs can bind such noncanonical sites with unexpectedly high affinity will require high-throughput target binding measurements for a larger set of miRNAs.

3. In Fig 1B, the authors showed binding activities of 6mer targets (better than 6mer-m8 and 6mer-A1), which is interesting to know if these sites are really functional in vivo. Can the authors do some sensor tests for these 6mer sites to see if they can mediate detectable regulation?

We agree that in vivo experiments in flies will be useful for the field, but they are beyond the scope of this paper and will need to await a future study.

4. The central paired sites are not identified in this study due to Ago1 mediated cleavage of these sites as shown in Fig 5b. Can these central paired sites (without seed match) really mediate detectable target cleavage in cells because it's presumably not involving GW182? Or these central paired sites can mediate target repression via normal miRNA way which is GW182 dependent? If these are real regulatory sites, it may encourage others to annotate such sites.

These experiments, which would require extensive (and expensive) degradome sequencing from flies, are best for future studies.

Minor concerns:

5. In Fig 1A, please indicate that the RNA oligo pool (input) is also sequenced.

We have added an arrow to the figure to indicate this.

6. In Fig 1A, the blocking oligonucleotide is DNA while the sequences are showed as RNA with "U".

We corrected U to T. Thank you!

7. In Fig 1C, there is a typo in X-axis-the last "100" should be "10,000".

We fixed the typo. Thank you!

8. In this sentence "In plants and animals, ~22-nt-long microRNAs (miRNAs) guide AGO-clade Argonaute proteins to repress partially complementary mRNA targets by accelerating their degradation 1–3 or inhibiting their translation 4–7." As known, in plants, miRNAs are most perfectly matched to target RNAs and guide their cleavage in an RNAi manner.

Tomari and co-workers reported in 2013 (Iwakawa et al., *Mol Cell* 2013; PMID 24267452) that *Arabidopsis* miRNAs can mediate translational repression without shortening the poly(A) tail or destabilizing the mRNA.

9. Please unify the citation format. “miRNAs display sequence-specific differences in binding their canonical sites: those with strong predicted free energy of site pairing bind their target sites with higher affinity than miRNAs with weak seed-pairing 21 ; McGeary et al., 2019, #16; Jouravleva et al., 2022, #143956}.”

Fixed.

10. Typos in these sentences: “To study the energetics of canonical binding of fly miRNAs, we loaded recombinant Ago1 loaded with one of four different miRNAs”; “Our data suggest that Ago1 may bind sites containing one single central mismatch but identifying universal rules for predicting which miRNAs can bind 8mer-x4R sites with unexpectedly high affinity high-throughput target binding measurements for a larger set of miRNAs”

We fixed the typos. Thank you!

Reviewer #3 (Remarks to the Author):

The manuscript by Vega-Badillo et al. describes the biochemical exploration of in vitro miRNA binding in the context of Drosophila Ago1. Building upon previous use of this technology for mammalian Ago2, the authors describe distinctions in Drosophila Ago1-mediated miRNA interactions that may help future predictive efforts in Drosophila. In general, the experiments are well-done and well-described, and generates useful insights for the fly miRNA community to further explore the mechanisms of how miRNAs regulate various aspects of biology in that system.

I only really have minor comments:

“At these positions mismatched nucleotides A and G are often the least tolerated, likely because their purine rings are larger than pyrimidines and thereby cause more steric hindrance.” – can the authors provide some quantification of “often”? It seems true for let-7, but somewhat less for the others – I think just having some number here would be clearer than a vague ‘often’

Thank you for noting this imprecision in the text. We edited the relevant sentence to read, “At these positions mismatched nucleotides A and G are the least tolerated in half of the occurrences, likely because their purine rings are larger than pyrimidines and thereby cause more steric hindrance.”

“Our data suggest that Ago1 may bind sites containing one single central mismatch but identifying universal rules for predicting which miRNAs can bind 8mer-x4R sites with unexpectedly high affinity high-throughput target binding measurements for a larger set of miRNAs.” – this sentence seems like it’s missing an ending (‘our data suggest ... but identifying universal rules for ... [will be impossible? Will require more extensive profiling?])’

We fixed the typo. Thank you!

The analysis in Fig 6 is interesting, but it might be good to show separately in supplementary figures the 5' and 3' end versions, since the 5' side is also impacted by whether the sequence base-pairs with the miRNA. I'm also a bit confused by the conclusion of the Fig 6 section – is the overall argument in this section that secondary structure has been shown to influence repression in [[emphasis mine]]mammalian]] cells, and this section validates the same principle for Drosophila Ago1? Or that RBNS scores can essentially serve as a functional proxy that correlates with (and thus likely incorporates the effect of) local structure? I don't think it's clear from the manuscript whether the same finding is true (& has already been shown) for RBNS AGO2 data in human, or just the general principle

We thank the reviewer for this thoughtful comment. McGeary and colleagues showed that flanking dinucleotide sequence context influences mammalian AGO2 binding in vitro (Fig. 4 in McGeary et al., *Science* 2019). Our earlier work further suggested that flanking dinucleotide sequence context affects how fast mammalian miRNAs associate with their targets (i.e., impacts k_{on}), likely because miRNA binding competes with local secondary structure of target RNA (Becker et al., *Molecular Cell* 2019; Jouravleva et al., *Cell Reports Methods* 2019). The results presented here indicate that the same principle also applies to *Drosophila* Ago1.

As suggested, we separately analyzed sites grouped by the dinucleotide sequences flanking either the 5' or the 3' ends of the 8mer seed. This divided the binding sites into 16 categories (4^2), and we quantified their K_D values using RBNS (Reviewers' Figure 2). However, reducing the number of sites from 256 categories (when considering both 5' and 3' flanks simultaneously) to 16 reduced statistical power, and no additional insights emerged from the separate analyses. We repeated this approach for 7mer m8, 7mer A1, 6mer, 6mer m8, and 6mer A1 sites across all five miRNAs tested, and reached the same conclusion.

I appreciate that the authors haven't tried to over-sell the impact of this work, but I think having a little more discussion of the impact of this work would be useful – I can see the obvious impact in aiding Drosophila-specific prediction tools, but I think having a touch more insight into what else the authors envision of the impact would be great

We would prefer to err on the side of under-selling. The miRNA binding field is already unnecessarily contentious (see the comments from Reviewer 1). We hesitate to add additional interpretations or claims to the manuscript.

a5'-site NN-3' $\rightarrow 2^4 = 16$ sequences
b

5'-NN site -3' → 2⁴ = 16 sequences

Reviewers' Figure 2 | The influence of flanking dinucleotide sequences. Relationship between the accessibility score and K_D values for the 16 sites containing one canonical target site flanked by different dinucleotide combinations (magenta NN in the schematic) at the 3' (a) or 5' (b) end of the binding site. Our analysis included 8mer, 7mer m8, 7mer A1, 6mer, 6mer m8, and 6mer A1 for all five miRNAs. r and p -values were calculated using Pearson correlation.

We thank the Reviewers for their interest in our work and their support. We address the remaining concerns of Reviewer #1 below.

Regarding the last part of major concern 5, I apologize that my initial comments were unclear, but I was requesting that the authors show the deltaG penalties calculated from the experimental values (as calculated from RBNS-derived Kd values by the equation $\Delta G = RT \ln(Kd)$), rather than the deltaG values predicted from the nearest-neighbor parameters. Showing the experimental values in Figure S4 should speak to the claim that fly Ago1 is less tolerant of seed-region imperfections than human AGO2.

We thank the reviewer for this clarification. In response, we plotted the ΔG penalties associated with individual seed-region imperfections across the five fly miRNAs examined in this study, as well as the mammalian miRNAs from previously published AGO2 RBNS datasets. The Reviewer's suggestion made us realize that, for the miRNAs shared between the fly and mammalian datasets—let-7 and miR-124—the ΔG of a perfectly matched g2–g8 target is lower in the mammalian datasets, while one would expect binding at 25°C to be more stable. This observation suggests that the reduced tolerance for mismatches by fly Ago1 does not arise because mismatches are inherently more destabilizing. Instead, the overall dynamic range of binding starts with weaker binding. Because flies have a lower body temperature than mammals, diffusion-limited association rates and target-search miRNA dynamics are slower. If the intrinsic biochemical properties of miRNA-loaded Ago1 were identical to those in mammals, fly Ago1 miRISCs would spend “too long” bound to suboptimal target sites.

We added the following text to the paragraph discussing the repertoire of fly Ago1 target sites:

“Our de novo site discovery algorithm identified only three binding sites for fly miR-124–loaded Ago1: the canonical 8mer, 7mer-m8, and 7mer-A1 (Fig. 5a). Remarkably, AGO2 guided by human miR-124, whose sequence is identical to that in flies, binds ~15 site types within a 10-fold range of the 8mer affinity, spanning canonical and noncanonical interactions, including seed-matched sites with wobble pairs, mismatches, bulges, and 3'-only sites²⁴.

Considering these differences in site recognition, we examined the ΔG penalties associated with individual seed-region imperfections across the five fly miRNAs examined in this study and the mammalian miRNAs from previously published AGO2 RBNS datasets^{24,25}. Binding affinity for a site is predicted to be more stable at 25°C than 37°C. Contrary to expectation, for miR-124 and let-7, which are shared between the fly and mammalian datasets, binding to a perfectly matched g2–g8 target was more stable for mammalian AGO2 at 37°C than for fly Ago1 at 25°C (Fig. S4). This observation suggests that the reduced tolerance for mismatches

of fly Ago1 does not arise because mismatches are inherently more destabilizing. Instead, the overall dynamic range of binding starts with weaker binding. We also note that our binding experiments were performed using 3.5 mM Mg²⁺. As expected, when the experiments were performed using 0.89 mM Mg²⁺, the concentration used previously to study binding of human AGO2 (Ref. ²⁴), we recovered fewer site-types (Fig. S5a). Together, our data suggest that fly Ago1 has lower tolerance for pairing imperfection because it binds perfect seed matches less tightly than AGO2.”

With respect to minor concern 1, the current model for miRNA-mediated repression centers on miRNA-mediated recruitment (through TNRC6) of a deadenylase complex (PAN2/PAN3 or CCR4-NOT) to accelerate mRNA poly(A) tail shortening, which in most settings enhances mRNA decay by hastening the time at which the tail becomes too short to resist mRNA decapping. In addition, the CCR4-NOT deadenylation complex brings to the mRNA the proteins responsible for translational inhibition of the mRNA (i.e., 4EHP and 4E-T; PMID28487484 from Sonenberg lab). Because the proteins that shorten the tail and the proteins that inhibit translation presumably arrive concurrently, it seems likely that the early step in mRNA decay (i.e., accelerated deadenylation) is happening in parallel with (and at the same time as) the effects on translation. These more recent mechanistic insights are not consistent with the earlier proposals that mRNA decay is a secondary effect of translational repression; for these earlier proposals to be plausible, translational repression would need to occur before accelerated poly(A) tail shortening (which does not appear to be the case in the settings examined). I hope this clarifies the concept that miRNAs mediate a combination of mRNA degradation and translational repression.

We thank the reviewer for this clarification. We edited our text to read,

“In plants and animals, ~22-nt microRNAs (miRNAs) guide AGO-clade Argonaute proteins to repress partially complementary mRNA targets by accelerating their degradation{Baek et al., 2008; Guo et al., 2010; Selbach et al., 2008} and inhibiting their translation, with recent mechanistic insights indicating that these effects may occur in parallel rather than as a sequential process {Chapat et al., 2017}.”

One more clarification: Regarding the last part of major concern 2 (which the authors found puzzling “because the original version of the manuscript stated that flies and mammals behaved similarly”), my concern was not with the quoted text but with a statement in the original version of the text, saying, “Unlike in mammals, these differences can be predicted by existing models of RNA duplex stability in solution.”

Understood, thank you.